# The genotype of barley cultivars influences multiple aspects of their associated microbiota via differential root exudate secretion

**Alba Pacheco-Moreno[1], Anita Bollmann-Giolai[1¤a], Govind Chandra[1], Paul Brett[1], Jack Davies[1¤b], Owen Thornton[1¤c], Philip Poole[2], Vinoy Ramachandran[2], James K. M. Brown[1], Paul Nicholson[1], Chris Ridout[1,3], Sarah DeVos[1,3], Jacob G. Malone**  [1,4]*

**1** John Innes Centre, Norwich Research Park, Colney Lane, Norwich, United Kingdom, **2** Department of Biology, University of Oxford, South Parks Road, Oxford, United Kingdom, **3** New Heritage Barley, Norwich Research Park, Norwich, United Kingdom, **4** School of Biological Sciences, University of East Anglia, Norwich, United Kingdom

¤a Current address: Max Planck Institute for Biology, Tübingen, Germany
¤b Current address: IBERS, Aberystwyth University, Aberystwyth, Ceredigion, United Kingdom
¤c Current address: Rothamsted Research, Harpenden, United Kingdom
* jacob.malone@jic.ac.uk

**Data Availability Statement:** Two data submissions have been made to ArrayExpress and are noted in the paper text: 1) Genome sequencing

## Abstract

Plant-associated microbes play vital roles in promoting plant growth and health, with plants secreting root exudates into the rhizosphere to attract beneficial microbes. Exudate composition defines the nature of microbial recruitment, with different plant species attracting distinct microbiota to enable optimal adaptation to the soil environment. To more closely examine the relationship between plant genotype and microbial recruitment, we analysed the rhizosphere microbiomes of landrace (Chevallier) and modern (NFC Tipple) barley (*Hordeum vulgare*) cultivars. Distinct differences were observed between the plant-associated microbiomes of the 2 cultivars, with the plant-growth promoting rhizobacterial genus *Pseudomonas* substantially more abundant in the Tipple rhizosphere. Striking differences were also observed between the phenotypes of recruited *Pseudomonas* populations, alongside distinct genotypic clustering by cultivar. Cultivar-driven *Pseudomonas* selection was driven by root exudate composition, with the greater abundance of hexose sugars secreted from Tipple roots attracting microbes better adapted to growth on these metabolites and vice versa. Cultivar-driven selection also operates at the molecular level, with both gene expression and the abundance of ecologically relevant loci differing between Tipple and Chevallier *Pseudomonas* isolates. Finally, cultivar-driven selection is important for plant health, with both cultivars showing a distinct preference for microbes selected by their genetic siblings in rhizosphere transplantation assays.

of 22 Pseudomonas isolates from the rhizosphere of the barley variety Chevallier and 20 Pseudomonas isolates from the rhizosphere of the barley variety Tipple. Accession number: E-MTAB-12917. 2) RNA-Seq of Pseudomonas fluorescens SBW25 isolated from axenic rhizospheres of the barley varieties Chevallier and Tipple one and five days post-inoculation. Accession number: E-MTAB-12918 All other relevant data are within the paper and its Supporting Information files.

**Funding:** APM was funded by UKRI-BBSRC DTP Studentship Award (BB/M011216/1) to the Norwich Research Park. ABG was supported by a John Innes Foundation Rotation PhD Studentship. GC, PB, JD, OT, JKMB, PN, CR and JGM were supported by BBSRC Institute Strategic Programme Grant BBS/E/J/000PR9797 to the John Innes Centre. PP and VR were supported by UKRI-BBSRC Responsive mode Grant BB/T001801/1 to PP. CR and SD were supported by UKRI-BBSRC Responsive mode Grant BB/K02003X/1 to CR. The funders played no role in the study design, data collection and analysis, decision to publish, or preparation of the manuscript.

**Competing interests:** The authors have declared that no competing interests exist.

**Abbreviations:** ASV, amplicon sequence variant; CER, controlled environment room; CFU, colony-forming unit; CRB, Congo red binding; dpi, days post-inoculation; FE, fluorescence emission; GC-MS, gas chromatography–mass spectrometry; gDNA, genomic DNA; HCN, hydrogen cyanide; LB, lysogeny broth; PA, protease activity; PBS, phosphate-buffered saline; PCoA, principal coordinates analysis; PGPR, plant-growth promoting rhizobacterium; RT, room temperature; TY, tryptone yeast; UMS, universal minimal salt; WT, wild-type.

## Introduction

Plants grow in close association with diverse bacteria, fungi, protozoa, archaea, and viruses that can influence the plant host in different ways by improving growth, protecting against pathogens, or conferring adaptive advantages [1–3]. Every plant species possesses a unique microbial signature, or microbiome, which extends their capacity to adapt to the surrounding environment and enhances their ability to withstand biotic and abiotic stresses [4–6]. Distinct microbial communities inhabit different niches within the plant such as the phylosphere, endosphere, and rhizosphere, with each of these micro-habitats providing a specific niche to which microbes must adapt and the composition of each compartment's microbiome differing within an individual plant [7].

The rhizosphere and roots are hotspots for microbial life because of the availability of plant-derived root exudates. Plants secrete an average of 21% of their photosynthates into the rhizosphere, which triggers a specialised shift in microbial composition and increases microbial density 10 to 1,000 times within the root influenced area [4,8–10]. The microbiome of the rhizosphere constitutes the most populous and diverse set of microorganisms directly interacting with the plant. Consequently, the impact of this community on plant health, disease, and productivity is highly significant [11].

Given that in general, soil nutrient content is quite low, access to plant-derived nutrients exacerbates the competition between microbial species for niche colonisation [12,13]. In response, bacteria have developed numerous ecological traits to improve their rhizosphere colonisation competitiveness [14]. Among these, metabolic versatility is particularly important in *Pseudomonas*, a ubiquitous and important bacterial genus that lives in a wide range of environments [15,16]. Plant-associated pseudomonads, such as the plant-growth promoting rhizobacterium (PGPR) *Pseudomonas fluorescens* nonspecifically colonise a vast range of different plants and exert important impacts on plant health by enhancing nutrient availability, suppressing pathogens, and priming the plant immune system [17]. *P. fluorescens* colonise diverse ecological niches and have correspondingly complex genomes, typically encoding approximately 6,000 genes with a remarkably high level of intraspecies diversity [18,19]. Much of the *P. fluorescens* accessory genome is devoted to signal transduction, environmental interactions, and specialised metabolism [17,18].

Researchers have recently begun to uncover the molecular mechanisms underpinning the relationship between plants and their rhizosphere microbiomes. This includes key concepts such as the importance of starting soil microbiome [20], founder effects [20,21] and local plant habitat [22,23] on community composition, and the importance of microbiome composition for plant health and pathogen biocontrol [6,24,25]. Furthermore, rhizosphere community assembly is actively shaped by plant genetic determinants. For example, Bulgarelli and colleagues showed that the barley genotype influences both root and rhizosphere microbiota, with variation seen for many microbes from diverse phyla [26]. Differential recruitment of taxonomically distinct rhizosphere bacteria was subsequently linked to a small number of loci in barley, with an NLR-like gene one of the main drivers of this phenomenon [27]. Another recent publication showed how a previously characterised receptor kinase, FERONIA [28] negatively modulates *P. fluorescens* colonisation of *Arabidopsis thaliana* by controlling the basal levels of reactive oxygen species [29].

Root exudate composition has also been shown to exert a significant effect on rhizosphere microbial recruitment. In *Avena barbata* bacterial soil isolates are differentially affected by plant growth, producing a split in the bacterial community based on their response to root growth over time [30]. Meanwhile, the secretion of coumarins [31] and root-derived triterpenes [32] in *A. thaliana* have been shown to play important roles in the rhizosphere

microbiome assembly process, with the latter exerting strong impacts on the abundance and composition of Bacteroidetes and δ-proteobacteria [32]. While our understanding of rhizosphere microbiome assembly is rapidly advancing, at this stage we still know little about the impact of plant genotype on selection within individual soil species. Given the striking differences in plant-beneficial phenotypes seen for different members of the same PGPR species complex [19,24], a better understanding of the principles underlying microbial genotype selection in the rhizosphere is a priority.

To address the effect that plant genotype exerts on microbial recruitment, we analysed the rhizosphere microbiomes of 2 distinct barley cultivars with markedly distinct histories and genetic backgrounds. Chevallier is an English landrace cultivar first selected in the 1820s before breeding programs were established, whereas NFC Tipple (hereafter "Tipple") is a modern cultivar released in 2004 by Syngenta Seeds [33]. We observed distinct differences in the abundance of rhizosphere and root-associated microbes between the 2 varieties, with Tipple recruiting significantly more *Pseudomonas* bacteria than Chevallier. This selection also manifested in distinct genotypic and phenotypic clustering within the recruited *Pseudomonas* populations. Cultivar-driven selection in barley appears to be strongly driven by root exudate composition, with the greater abundance of hexose sugars secreted from Tipple roots selecting *Pseudomonas* isolates adapted to grow on these carbon sources. Cultivar-driven selection was also observed for individual loci; both at a population level and in differential gene transcription in the model organism *P. fluorescens* SBW25. Exudate-driven microbial selection appears to confer growth advantages on barley plants, with both cultivars showing a preference for their native microbiota, supporting a link between cultivar-driven rhizosphere selection and plant health.

## Results

### Barley cultivar impacts root-associated microbial communities

To gain an overview of the microbiome structure of the Chevallier and Tipple root systems, we used targeted amplicon sequencing of the 16S rRNA gene and ITS regions from gDNA samples taken from three-week-old greenhouse barley plants grown in JIC Cereal Mix compost (Figs 1 and S1, and S2). Cereal Mix was chosen as a growth substrate for our greenhouse experiments to enable consistency across the extended period of this study. While the soil in this compost is sterilised to kill soil-borne cereal pathogens, Cereal Mix nonetheless retains a complex microbiome derived from the peat component. The relative abundance of amplicon sequence variants (ASVs) [34] for the 30 most represented bacterial genera are shown in Fig 1A. Overall, 63.3% of all ASVs were Proteobacteria, 26.7% Actinobacteria, 6.7% Acidobacteria, and 3.3% Bacteroidetes.

In the bulk soil, the most abundant genera were *Rhodanobacter* (8.35%), *Acidothermus* (6.82%), and *Conexibacter* (5.11%) (Fig 1A). The rhizosphere of the 2 cultivars (adonis test, *p*-value 0.001) and the different plant compartments (ANOSIM test, *p*-value 0.001) differed significantly in their bacterial composition. In the Tipple rhizosphere, *Pseudomonas* (7.55%) and *Massilia* (6.62%) presented the greatest relative abundance, while in the Chevallier rhizosphere, the most abundant genera were *Massilia* (7.67%) and *Acidothermus* (5.65%), with *Pseudomonas* representing 4.87% of total ASVs. In the Tipple root compartment, *Pseudomonas* (13.43%), *Klebsiella* (8.82%), and *Massilia* (8.36%) were highly abundant, while for Chevallier, the most abundant genera were *Massilia* (17.03%), *Pseudomonas* (7.15%), members of the Burkholderiaceae family (7.04%), and *Klebsiella* (6.49%) (Fig 1A).

Next, principal coordinates analysis (PCoA) using Bray–Curtis distance measurement was used as a measure of the variation of taxonomic profiles between samples (Fig 1B). The first

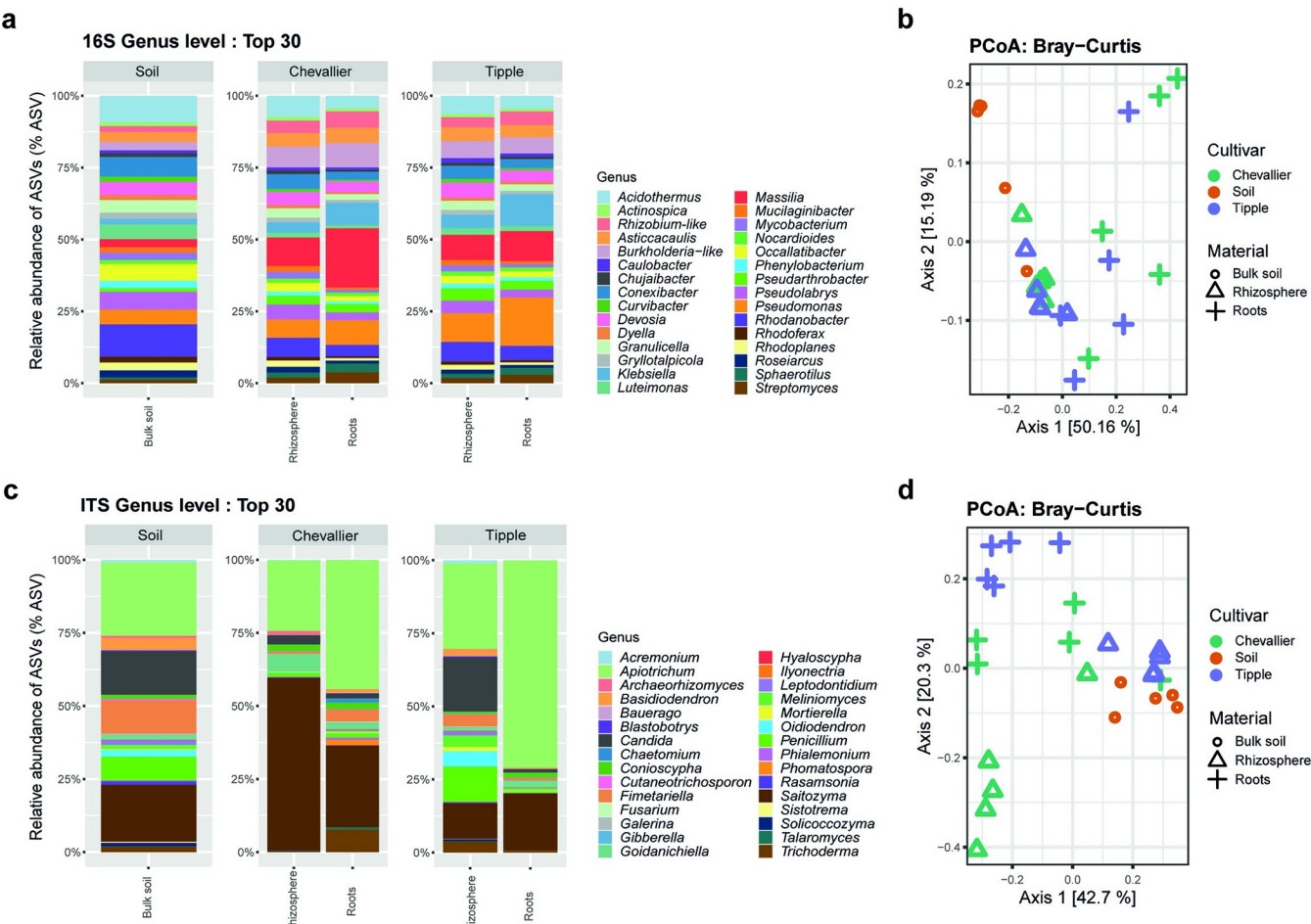

**Fig 1.** Metabarcoding analysis of the microbial communities associated with Chevallier, Tipple, and bulk soil. (A) Top 30 bacterial ASV community composition. (B) PCoA showing beta-diversity with Bray–Curtis distance for bacterial community composition. (C) Top 30 fungal ASV community composition. (D) PCoA showing beta-diversity with Bray–Curtis distance for fungal community composition. Two plant compartments, root endosphere and rhizosphere, and the bulk soil were analysed. Five replicates were used per condition. In PCoA, green represents Chevallier, orange, bulk soil and purple, Tipple. Black hollow symbols represent plant compartment, round is bulk soil; triangle is rhizosphere and cross represents roots or endosphere. The data underlying this figure can be found in S1 Data. ASV, amplicon sequence variant; PCoA, principal coordinates analysis.

component accounted for about 50% of the variance, with samples clustering both by compartment (rhizosphere, root, or bulk soil) and host cultivar. Bacterial composition between cultivars was significantly different for whole plants (adonis test, *p*-value 0.001) as well as between roots, rhizosphere, and bulk soil (ANOSIM test, *p*-value 0.001). Within individual cultivars, both the Chevallier (ANOSIM test, *p*-value 0.01) and Tipple (ANOSIM test, *p*-value 0.01) root and rhizosphere communities were statistically distinct from one another. Finally, the Chevallier and Tipple rhizosphere communities were also significantly different (adonis test, *p*-value 0.01).

*Saitozyma* and *Apiotrichum* were the most abundant plant-associated fungal genera, together accounting for up to 90% of the ASV abundance in some samples (Fig 1C). These 2 genera were also abundant in bulk soil, with *Candida* (14.78%), *Fimetariella* (11.40%), and *Penicillium* (8.17%) the next most prevalent genera. The Chevallier rhizosphere community differed markedly from bulk soil, with a very high abundance of *Saitozyma* and *Goidanichiella* (5.78%). By contrast, the Tipple rhizosphere displayed a very similar profile to bulk soil, with

*Candida* (18.39%) and *Penicillium* (12.01%) among the most abundant genera. The Tipple root compartment displayed a dramatic reduction in fungal diversity, with most samples overtaken by *Apiotrichum* and to a lesser extent *Saitozyma*. This reduction in diversity was less marked for the Chevalier root system. Here, *Trichoderma* (7.66%) was the third most abundant genus in the root compartment. Bray–Curtis beta-diversity showed clear clustering of samples both by host cultivar and by compartment (Fig 1D), with significant differences observed between both host cultivars (ANOSIM test, *p*-value 0.005) and compartments (adonis test, *p*-value 0.002).

To assess the species richness across barley cultivars and plant compartments, we calculated alpha-diversity for bacterial and fungal communities (S1 and S2 Figs). For the bacterial population, alpha diversity decreased progressively from bulk soil to rhizosphere to the root compartment. As shown by Shannon index, significant differences were observed between the rhizospheres of both cultivars and the bulk soil, but not between the 2 rhizosphere communities (S1A Fig). Differences in the observed species richness (ANOVA test, *p*-value 0.045, df (degrees of freedom; first value—degrees of freedom for the independent variable; second value—degrees of freedom for the residuals) = 1,8, F value 5.598) were seen for the root endosphere communities of the 2 cultivars (S1B Fig), but not for Shannon index. For Chevallier, both observed richness (Kruskal–Wallis test, *p*-value 0.03) and Shannon index (Kruskal–Wallis test, *p*-value 0.009) were significantly different between plant compartments (S1C Fig), while Tipple presented a marked difference between rhizosphere and root bacterial communities by Shannon index (Kruskal–Wallis test, *p*-value 0.009) but not observed richness (S1D Fig).

Alpha diversities of the fungal population were significantly different between the Chevallier rhizosphere and soil for both observed richness (ANOVA test, *p*-value $9.6E^{-5}$, df = 2,12, F value = 22.02) and Shannon index (pairwise Wilcox test, *p*-value 0.024) and between Chevallier and Tipple rhizospheres (observed, ANOVA test, *p*-value 0.0024, df = 1,8, F value = 18.95; Shannon, ANOVA test, *p*-value 0.0077, df = 1,8, F value = 20.04). No other significant differences were observed, (S2B and S2C Fig), apart from a highly significant change in alpha-diversity between the Tipple root and rhizosphere populations (observed, ANOVA test, *p*-value 0.004, df = 1,8, F value = 15.33; Shannon, ANOVA test *p*-value 0.0002, df = 1,8, F value = 39.92) (S2D Fig).

These experiments suggest that (i) ecological niche plays a fundamental role in microbial community assembly and structure; (ii) species richness progressively decreases from the soil to the root endosphere; (iii) barley genotype affects the assembly of plant-interacting microbial consortia; (iv) Chevallier substantially modifies the fungal composition of its rhizosphere, while Tipple does not; and (v) key bacterial genera are differentially enriched by Tipple (e.g., *Pseudomonas*) or Chevallier (e.g., *Massilia*) root environments.

## Rhizosphere *Pseudomonas* genotypes cluster according to plant genotype

To interrogate the cultivar-specific differences more closely, we next examined the *Pseudomonas* populations associated with Chevallier and Tipple. While several fungal and bacterial genera displayed larger cultivar-specific differences, *Pseudomonas* spp. have important effects on plant growth and health, are easy to isolate and are genetically tractable, making them a good model for downstream molecular analysis. We used CFC selective agar to randomly isolate 239 bacterial strains from the roots of our initial 3-week-old barley plants, alongside 31 bulk soil isolates. Genotyping based on the *gyrB* housekeeping gene [35,36] identified most isolates as *Pseudomonas* spp., predominantly from the *P. fluorescens* complex (S1 Table and Fig 2), and 21 isolates were determined to be microorganisms other than *Pseudomonas* and were excluded

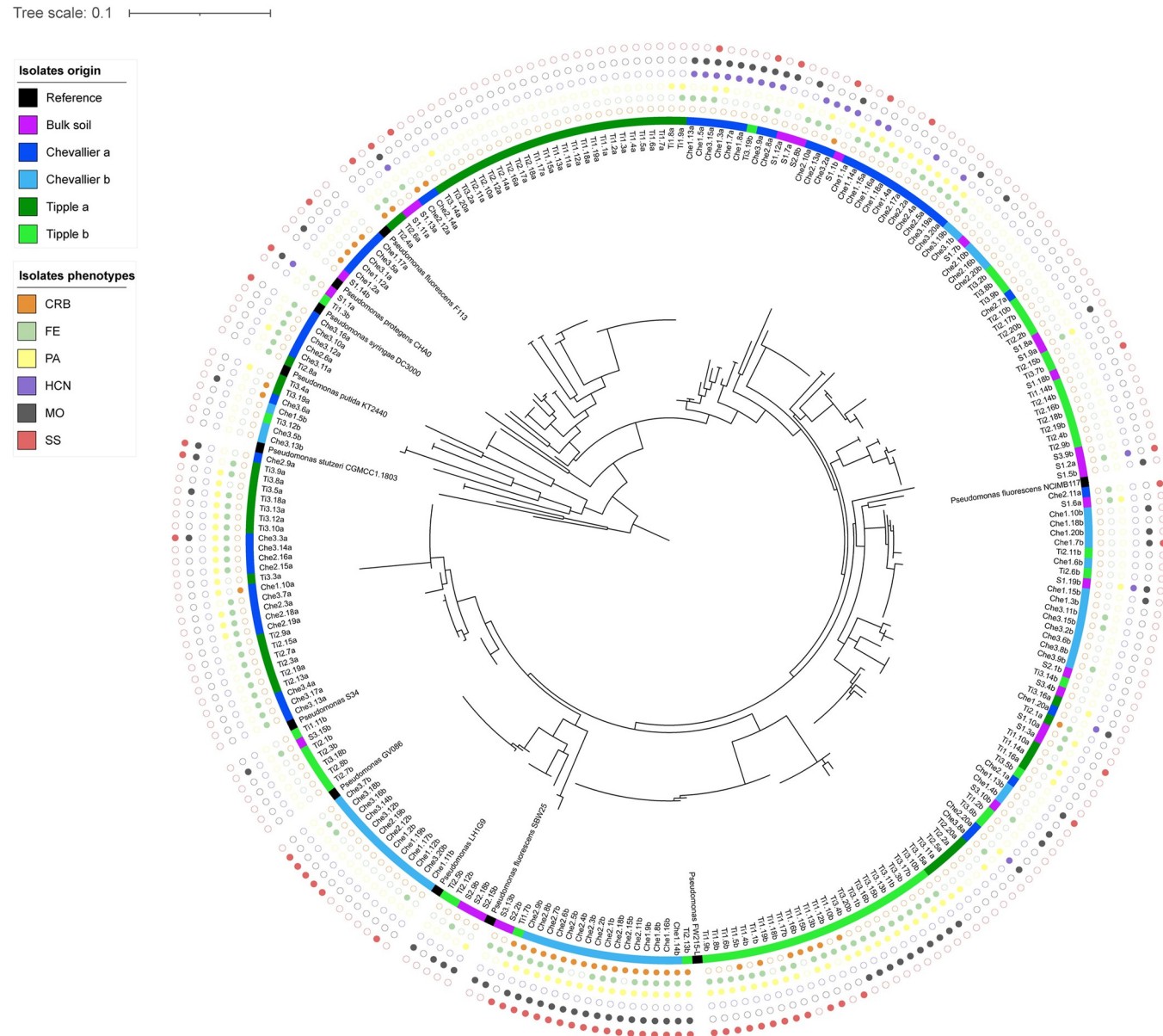

**Fig 2. *gyrB* phylogeny of the Chevallier and Tipple rhizosphere *Pseudomonas* isolates.** Two independent datasets, denoted with suffixes a and b, are included in the analysis. The phylogenetic tree is based on 260 partial sequences of the *gyrB* gene from sequenced and type strains and was constructed using the ML method and Tamura–Nei model with 1,000 bootstrap value. The isolation origin of samples is indicated by the coloured ring closest to the tree. Reference strains are based on publicly available genome sequences. Filled/empty circles in the outer rings denote the presence/absence respectively of the ecological traits, CRB = Congo red binding, FE = fluorescence emission, PA = protease activity, HCN = hydrogen cyanide, MO = motility and SS = *Streptomyces* suppression. The data underlying this figure can be found in S2 Data.

from further analysis. The resulting phylogenetic tree showed clear patterns of clustering, corresponding both to the experiment (and hence soil batch) but also to barley cultivar. The bulk soil isolates (representing a subset of the initial *Pseudomonas* population before barley planting) were distributed evenly throughout the tree. We then assessed the overall diversity present in the *Pseudomonas* population by computing the estimated divergence in *gyrB* sequences using the Tamura–Nei model [37]. By this calculation, the bulk soil isolates showed the greatest sequence divergence (0.1509), followed by the Chevallier isolates (0.1423), with the Tipple

population showing the least sequence divergence (0.1225). The total population had a distance index of 0.1371.

We next examined the extent to which the apparent genotypic selection within the rhizosphere *Pseudomonas* community translated into differences in the distribution of phenotypic traits. Isolates were tested for Congo red binding (CRB)—used here as a proxy for their biofilm forming capacity—, fluorescence emission (FE)—to measure siderophore production—, protease activity (PA), hydrogen cyanide (HCN) production, swarming motility (MO), and suppression of the Actinomycete *Streptomyces venezuelae* (SS). Ordinal values were then assigned to each trait (Fig 2). The Chevallier isolates contained a higher percentage of isolates with high CRB (Chevallier 27/120, Tipple 13/120, Soil 2/72), MO (Chevallier 42/120, Tipple 20/120, Soil 20/72), or HCN (Chevallier 21/120, Tipple 1/120, Soil 9/72) scores, whereas Tipple isolates scored lower for almost all the tested traits (S3 Fig). A striking degree of clustering was observed between phenotypic traits and the distribution of *Pseudomonas* genotypes in our phylogenetic tree (Fig 2).

## Chevallier and Tipple differ in root exudates composition

We hypothesised that the differences observed in microbiome composition and microbial genotype distribution between Chevallier and Tipple could be driven by differences in the secretion of root exudates. To test this, we used luminescent biosensors [38,39] to examine the secretion of specific groups of metabolites from the roots of barley seedlings grown in axenic microcosms. Fructose, C4-dicarboxylates, sucrose, and phenylalanine biosensors were selected to represent key groups of molecules often found in the rhizosphere; sugars, organic acids and amino acids [40], and plants were evaluated at 2, 5, and 7 days post-inoculation (dpi). Significant differences between cultivars were seen for the fructose biosensor, which responds to several hexose sugars [38] (S4 Fig), with especially high expression in Tipple at 2 dpi (Fig 3A).

We then explored the root exudates diversity directly using gas chromatography–mass spectrometry (GC-MS). Chevallier and Tipple barley plants were grown for 3 weeks under axenic conditions and their root exudates were extracted and analysed. Data for all samples was normalised and the median was established as a baseline for their abundance (peak intensity). Comparing the relative increase/decrease for each compound, a total of 105 entities with a $\log_2$-fold change > 2 were detected by GC/MS (Fig 3B and S2 Table). Compounds 1–59 showed a match in at least one of the compound libraries used, whereas 60 to 105 could not be identified. A two-way ANOVA indicated that plant genotype exerted a profound effect on the observed variation in the data, accounting for 4.98% of the total variation ($p$ <0.0001). Overall, a greater abundance of the unidentified metabolites was detected in the Chevallier exudates compared to Tipple. Two striking differences were seen between the 2 cultivars. First, a derivative of phosphoric acid was massively abundant in Chevallier exudates ($\log_2$-fold change 16.474) and absent in Tipple ($\log_2$-fold change −12.613). Second, D-glucose, whose $\log_2$ abundance was 4.389 in Tipple, was much less abundant (−2.405) in Chevallier exudates.

Overall, these data suggest that the Tipple rhizosphere is enriched in $C_6/C_{12}$ sugars, such as fructose and glucose. In contrast, Chevallier exudates contain a more diverse exudate composition and a greater abundance of unidentified molecules.

## Primary carbon metabolism is central to *Pseudomonas* spp. cultivar adaptation

To examine the link between the exudate profiles of Chevallier and Tipple and the rhizosphere *Pseudomonas* population, we tested the ability of the *Pseudomonas* isolate library to grow on different carbon sources. Comparing the final $OD_{600}$ values for 48-h grown cultures, we

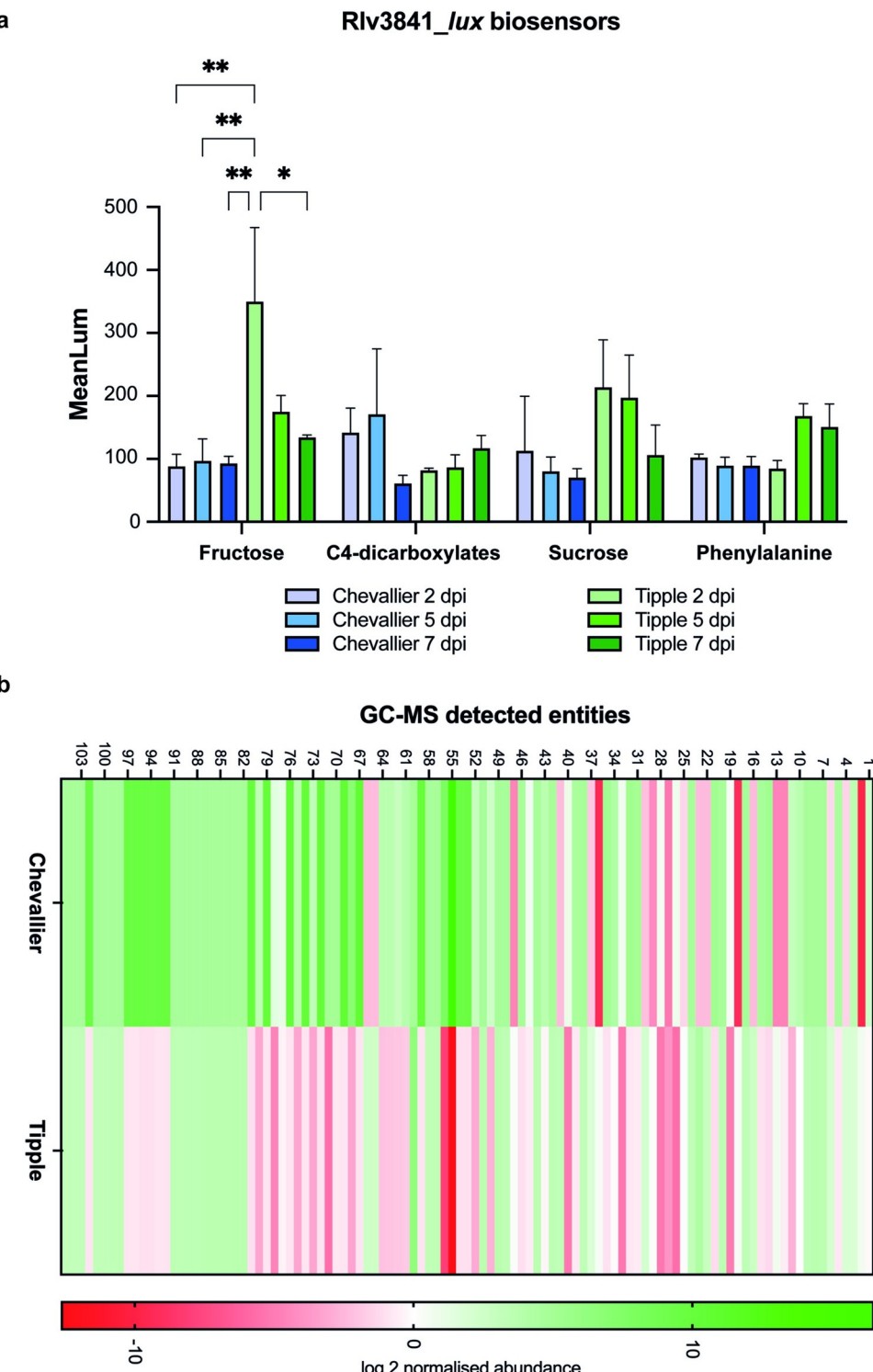

**Fig 3. Root exudate analysis of Chevallier and Tipple.** (A) In planta temporal screening of root metabolites produced by Chevallier and Tipple seedlings with luminescent biosensors. Fructose, C4-dicarboxylates, sucrose, and phenylalanine biosensors based on wild-type (WT) Rlv3841 [38] were inoculated in the roots of 2 days old Chevallier and Tipple seedlings. Images were acquired at 2, 5, and 7 dpi and 3 plants were used per condition. Mean values of luminescence intensity are shown in the graph. Error bars represent standard error of the mean, $p$-values were calculated by Tukey's multiple comparison test, and asterisks indicate $p < 0.05$ (*), 0.01 (**). A representative graph is shown from 3 independent experiments. (B) Chevallier and Tipple root exudates metabolites detected by GC-MS. The

heat map shows the overall composition of root exudates identified metabolites presenting a log2-fold change $\geq 2$ between the cultivars. The data underlying this figure can be found in S3 Data. dpi, days post-inoculation; GC-MS, gas chromatography–mass spectrometry.

observed 2 intriguing trends (Fig 4). When grown on acetate as the sole carbon source (Fig 4A), 9/60 Chevallier but only 3/60 Tipple isolates attained a final $OD_{600}$ >0.2796 (the 90th percentile value). Conversely, when grown on glucose (Fig 4B), 10/60 Tipple isolates reached a final $OD_{600}$ >0.2028 (the 90th percentile), but only 2/60 Chevallier isolates, supporting the hypothesis that root exudate differences may underpin the observed selection of *Pseudomonas* genotypes (Fig 2).

To investigate further, a subset of 42 isolates was selected for more detailed analysis, and 19 strains were chosen based on a combination of growth characteristics (Fig 4A and 4B) and phylogenetic distribution to ensure maximum genetic diversity (Fig 2). A further 23 isolates were then selected at random to reduce bias within the sample set. These isolates were again interrogated for growth in defined media. Alongside acetate, we selected fructose as a hexose sugar abundant in many root exudates [41,42], including Tipple. No obvious differences in growth rates were observed between Tipple or Chevallier isolates when grown on acetate (Fig 4C). However, when the isolates were grown on fructose (Fig 4D), a clear difference emerged between the Tipple and Chevallier isolates, with the Tipple-derived community growing faster and to a higher cell density than strains isolated from Chevallier.

Next, we produced a set of agar plates containing Chevallier and Tipple root exudates and observed the growth of our 42 sequenced isolates on these (S5 Fig). While several strains were able to grow to some extent on both exudates, more Chevallier isolates (9/22) grew on Chevallier root exudates than on Tipple (7/22), and vice versa (9/20 Tipple isolates grew on Tipple exudates and 6/20 grew on Chevallier exudates).

Finally, to study the effect of differential root exudation on microbial colonisation, we performed rhizosphere competition assays using Tipple and Chevallier, and 2 well-characterised metabolic mutants of *P. fluorescens* SBW25: Δ*rccR* and Δ*hexR* [43] in vermiculite microcosms. Δ*rccR* displays a pronounced growth defect on compounds that enter the Krebs Cycle via the Entner Doudoroff (ED) pathway. Conversely, Δ*hexR* struggles to grow on Krebs Cycle intermediates and two-carbon molecules such as acetate ([43], Fig 4D). While the Δ*rccR* mutant was less competitive than WT in both cultivars, its competitive disadvantage was significantly greater in the Tipple rhizosphere. Δ*rccR* and Δ*hexR* showed similar fitness in the Chevallier rhizosphere, but Δ*rccR* was strongly outcompeted by Δ*hexR* in Tipple (Fig 4E). Together, these data support the hypothesis that barley cultivars select *Pseudomonas* genotypes that can most effectively metabolise their root exudates.

## Different barley rhizospheres influence the abundance of specific bacterial loci

Next, we examined the effect of cultivar-specific selection on the genomes of the rhizosphere *Pseudomonas* community. Our 42 selected isolates were whole genome sequenced and reciprocal BLAST-searched for the presence of loci potentially involved in plant colonisation, using the model organism SBW25 as a reference genome. In the first instance, this included 410 previously described plant-induced *Pseudomonas* genes, encoding transporters, biofilm regulators, chemotaxis proteins, or siderophore synthases [44] alongside substrate transporters and transcriptional regulators—2 important traits relevant to niche adaptation [13,43] (see Materials and methods). Using a Chi-Square analysis, we identified 51 genes whose distributions significantly differed in the Chevallier and Tipple genomes (Fig 5A). Some genes were abundant

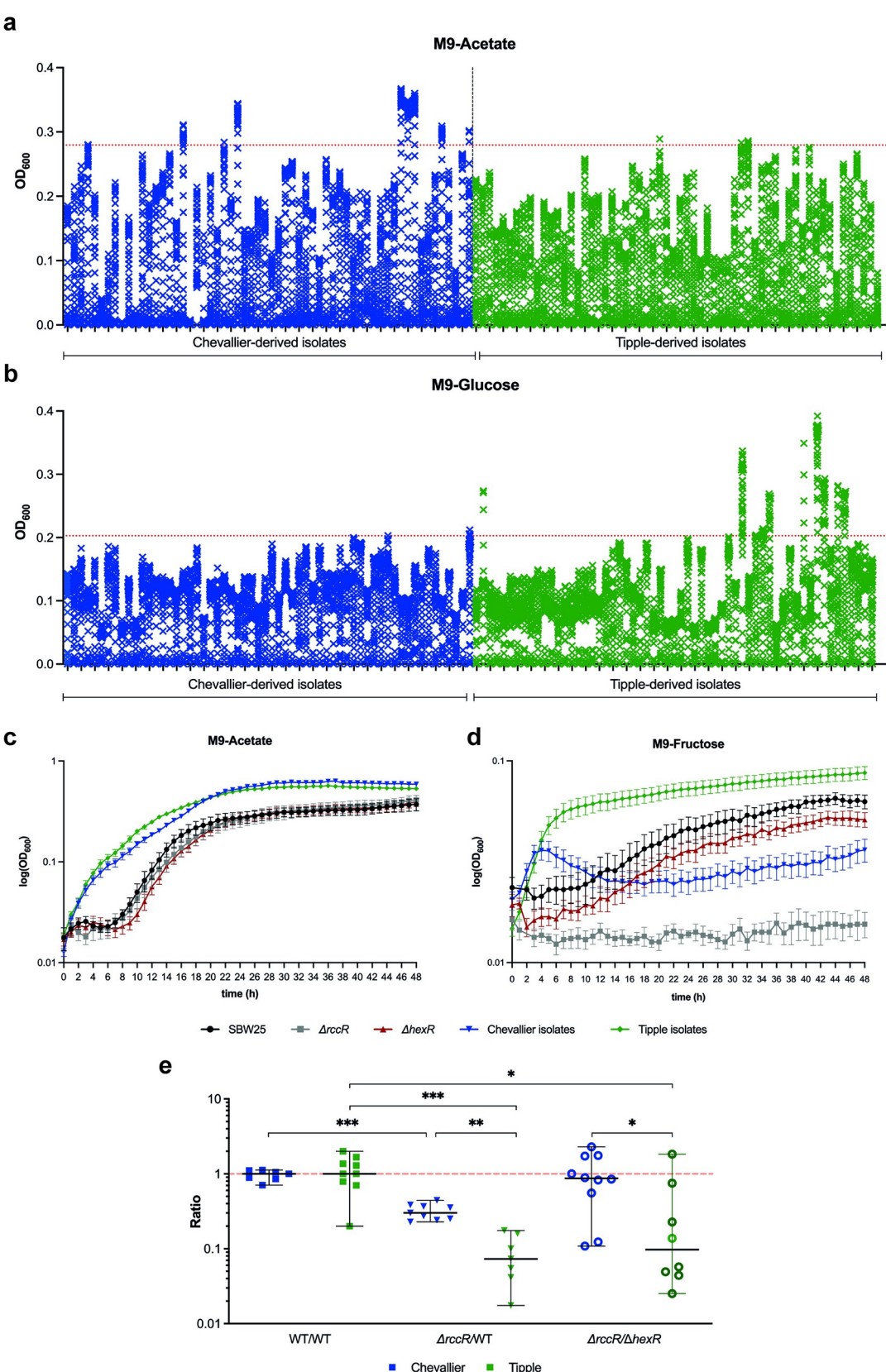

**Fig 4. Carbon regulation is a key determinant for rhizosphere microbe selection.** (A, B) Rhizosphere *Pseudomonas* isolate growth over 48 h in M9 medium supplemented with 0.4% Acetate (A) or 0.4% Glucose (B). Each cross in a vertical stack denotes the $OD_{600}$ value at sequential time points from 0 to 48 h. Values were normalised to the lowest value of each dataset. Chevallier isolates are coloured blue and Tipple isolates green. The red dashed line represents the 90th percentile of all 48 h $OD_{600}$ values. (C, D) Growth curves for the 42 rhizosphere *Pseudomonas* isolates selected for whole genome sequencing over 48 h in M9 medium supplemented with 0.4% acetate (C) or 0.4% fructose (D). SBW25 WT is shown in black, SBW25 Δ*rccR* in grey, SBW25 Δ*hexR* in red, pooled Chevallier isolate values in blue, and pooled Tipple isolates in green. Values for Chevallier and Tipple isolates are represented as the mean values of all the isolates tested. Std. errors shown in each case. (E) Chevallier/ Tipple rhizosphere colonisation competition assays. The graph shows the ratios of SBW25 WT and Δ*rccR* to SBW25-*lacZ* or SBW25 Δ*rccR-lacZ* to Δ*hexR*. Colony-forming units (CFUs) recovered from the rhizospheres of Chevallier (blue) and Tipple (green) barley cultivars at 5 dpi. Each dot represents the ratio of CFUs recovered from an individual plant, and 8–10 plants were used per condition and *p*-values were calculated by Mann–Whitney U test, asterisks indicate $p < 0.05$ (\*), 0.01 (\*\*), or 0.001 (\*\*\*). Experiments were repeated at least twice, and representative graphs are shown here. The data underlying this figure can be found in S4 Data. dpi, days post-inoculation.

in Tipple isolates but scarce in Chevallier strains, e.g., the sugar ABC transporter gene *PFLU_2583*. Likewise, others were found at higher rates in Chevallier isolates, such as *PFLU_0315*, encoding a GABA transporter.

To investigate the biological roles of the potential cultivar-selected genes, we produced non-polar deletions of 8 genes in SBW25. Genes were selected based on their predicted biological role and the extent that their frequency differed between the 2 cultivars. The deleted genes and their predicted functions are listed in Table 1. Growth assays in complex (KB and LB) and defined minimal media showed that Δ*6072* exhibited compromised growth in pyruvate, glucose, or glycerol minimal media (S6 Fig) implicating this predicted LysR-family regulator gene in the control of primary carbon metabolism.

Chevallier and Tipple rhizosphere competition assays were then carried out for the 8 mutants (Fig 5B). Δ*6072* showed an impaired colonisation phenotype versus WT SBW25 in both cultivars, which was highly significant in the Tipple rhizosphere. Given the growth impairment of this mutant on glucose, this result aligns with the observation that the Tipple rhizosphere is enriched in glucose and similar molecules. Δ*2583* also exhibited a fitness penalty, but only on Tipple plants. As mentioned above, *PFLU_2583* encodes a putative sugar transporter and is found in greater abundance in Tipple rather than Chevallier-associated isolates. Finally, Δ*5080* also displayed a compromised colonisation ability in Tipple rhizospheres only. These results support the idea that specific genetic traits are selected by root exudates in a cultivar-specific manner, and in turn, support rhizosphere colonisation on those specific cultivars.

## Cultivar-specific genes were differentially expressed in SBW25

Next, we investigated to what extent this genetic selection overlaps with differential gene expression in the barley rhizosphere. To address this, SBW25 mRNA abundance was studied by RNA-seq in the Tipple and Chevallier rhizospheres, following 1 and 5 days growth in vermiculite microcosms (Fig 6A–6F). A total of 2,158 genes were differentially expressed ($log_2$-fold $\leq$ -1 or $\geq$1 and *p*-value $\leq$ 0.05) compared to the cell-culture control in the rhizosphere of Chevallier at 1 dpi (Fig 6A), increasing slightly to 2,515 at 5 dpi (Fig 6B), and 2,039 genes were differentially expressed in the Tipple rhizosphere at 1 dpi (Fig 6C), of which 950 were up-regulated and 1,089 were down-regulated. At 5 dpi, this increased to 2,470 genes (Fig 6D).

In contrast to the high number of rhizosphere-regulated genes, pairwise comparisons between the Chevallier and Tipple rhizosphere samples (Fig 6E and 6F) showed a striking similarity in SBW25 gene expression. At 1 dpi, just 14 genes were differentially expressed between the 2 cultivars, with only 7 genes presenting a significant fold-change after 5 dpi (Fig 6G). Genes whose mRNA abundance significantly differed between the Tipple and Chevallier

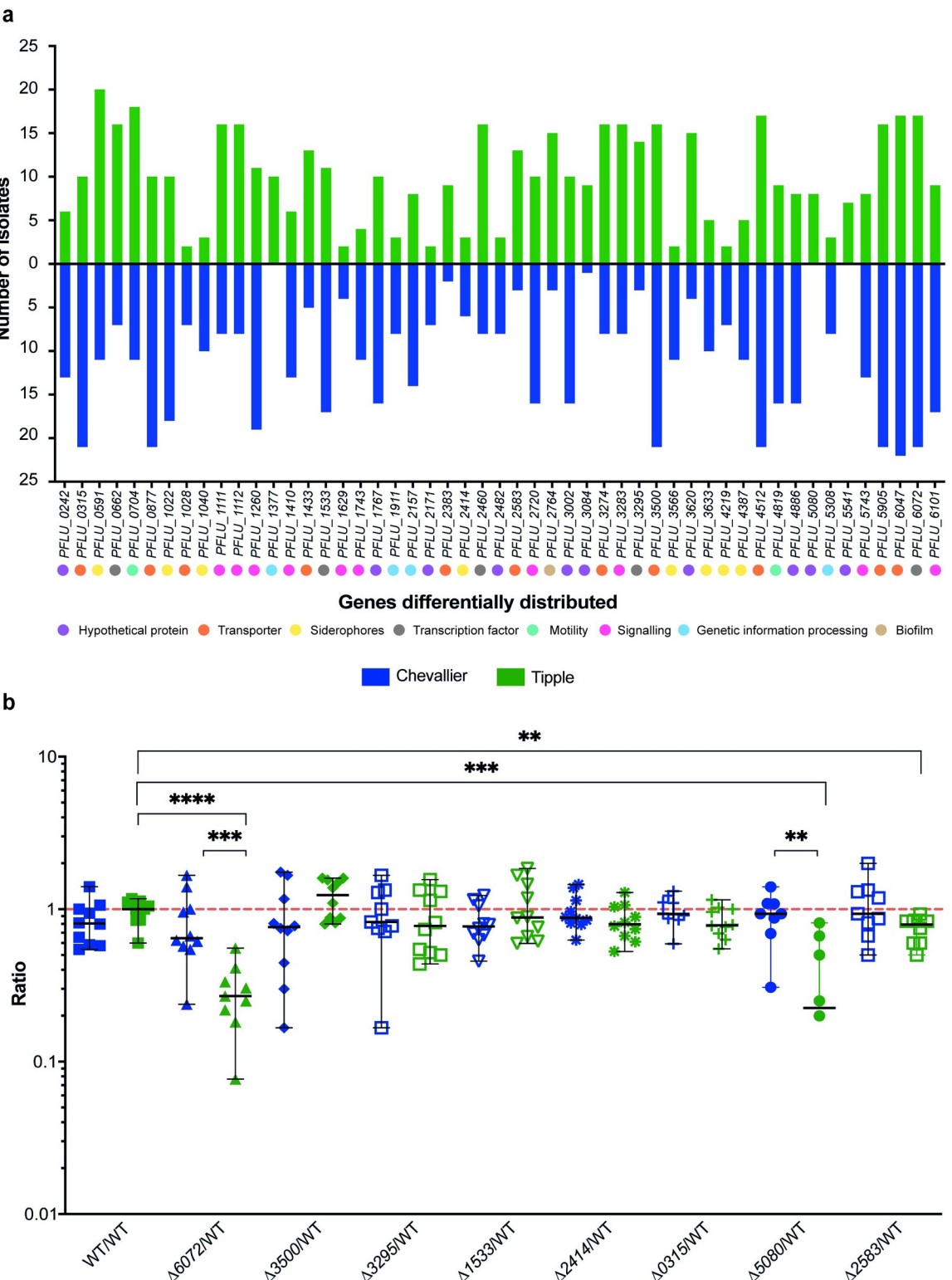

**Fig 5. Cultivar-driven gene selection within the barley rhizosphere *Pseudomonas* population.** (A) Distribution of genes differentially present in the sequenced *Pseudomonas* isolates from Chevallier and Tipple. The bars show the number of strains within the sequenced population where reciprocal BLAST suggests a particular gene is present. The SBW25 gene designation for each locus is given on the X-axis, with predicted gene functions denoted by coloured dots. Chevallier isolates are shown in blue and Tipple in green. All the genes represented here were significantly differently abundant ($p < 0.05$) between the 2 cultivars according to a Chi-Square test. (B)

Rhizosphere colonisation competition assay. The graph shows the ratios of differentially distributed mutants to SBW25-*lacZ* CFUs recovered from the rhizospheres of Chevallier (blue) and Tipple (green) barley cultivars. Each dot represents the ratio of CFUs recovered from an individual plant; *p*-values were calculated by Mann–Whitney U test and asterisks indicate $p < 0.05$ (*), 0.01 (**), 0.001 (***), or 0.0001 (****). The experiment was repeated 3 times and a representative graph is shown here. The data underlying this figure can be found in S5 Data.

rhizospheres are listed in Table 2. Curiously, most Tipple up-regulated genes were in the same gene cluster, despite forming at least 4 distinct operons (S7 Fig). No significant differences in gene expression were observed for the differently selected genes discussed above (Fig 5 and S3 Table). However, half of these loci were shown to be up-regulated in the rhizosphere relative to liquid culture, independent of cultivar (S4 Table).

Two of the most differently regulated genes in the rhizospheres of the 2 cultivars, *PFLU_3091* and *PFLU_4463*, were selected for further characterisation. The predicted amino acid permease gene *PFLU_3091* is highly up-regulated in the Tipple rhizosphere compared to Chevallier, while the EamA domain-containing gene *PFLU_4463* is down-regulated in Tipple. Non-polar deletions of both genes were produced in SBW25, and the relative competitiveness of the 2 mutants assessed in the Tipple and Chevallier rhizospheres (Fig 6H). The *Δ4463* mutant was strongly compromised for Chevallier rhizosphere colonisation, while the *Δ3091* strain was less strongly, but still significantly compromised, in the Tipple rhizosphere. These results support roles for both *PFLU_3091* and *PFLU_4463* in cultivar-specific colonisation of the barley rhizosphere by *P. fluorescens*.

## Rhizosphere selected microbes induce cultivar and microbiome-specific plant growth

Finally, to examine the consequences of microbial selection for plant growth and health, we conducted a rhizosphere cross-inoculation experiment (Fig 7). Chevallier and Tipple seedlings were axenically inoculated with either a Chevallier rhizosphere microbiome extract (CRh) or a Tipple rhizosphere extract (TRh) and with synthetic communities (SynComs) of Chevallier (CPs) or Tipple-derived *Pseudomonas* (TPs), then their dry weights were recorded after 3 weeks of growth. Tipple plants showed a small reduction in dry mass of −7 +/− 7.534% when inoculated with CRh (TxCRh) in comparison to a Tipple uninoculated control, against a slight increase in mass (+12 +/ 12.610%) when grown with their native rhizosphere community (TxTRh). Conversely, Chevallier showed strong positive responses to the rhizosphere extracts of both cultivars, with a mild preference for its own microbiome. Similarly, Chevallier inoculated with either CPs or TPs showed a growth increase (+15 +/− 12.14% and +20 +/− 9.806% dry mass, respectively), Tipple responded negatively to both SynComs. Together, these results suggest that Tipple is less capable than Chevallier of utilising a non-host rhizosphere

**Table 1. Cultivar-selected genes deleted in SBW25.**

| Gene ID | Predicted function of encoded protein | More abundant in: |
|---|---|---|
| *PFLU_0315* | GABA permease: *gabP* | Chevallier |
| *PFLU_1533* | LysR-family transcriptional regulator | Chevallier |
| *PFLU_2414* | Iron siderophore sensor | Chevallier |
| *PFLU_6072* | LysR-family transcriptional regulator | Chevallier |
| *PFLU_2583* | Putative rhizopine-binding protein | Tipple |
| *PFLU_3295* | GntR family transcriptional regulato*r: vanR* | Tipple |
| *PFLU_3500* | C4-dicarboxylate transporter | Tipple |
| *PFLU_5080* | Hypothetical protein | Tipple |

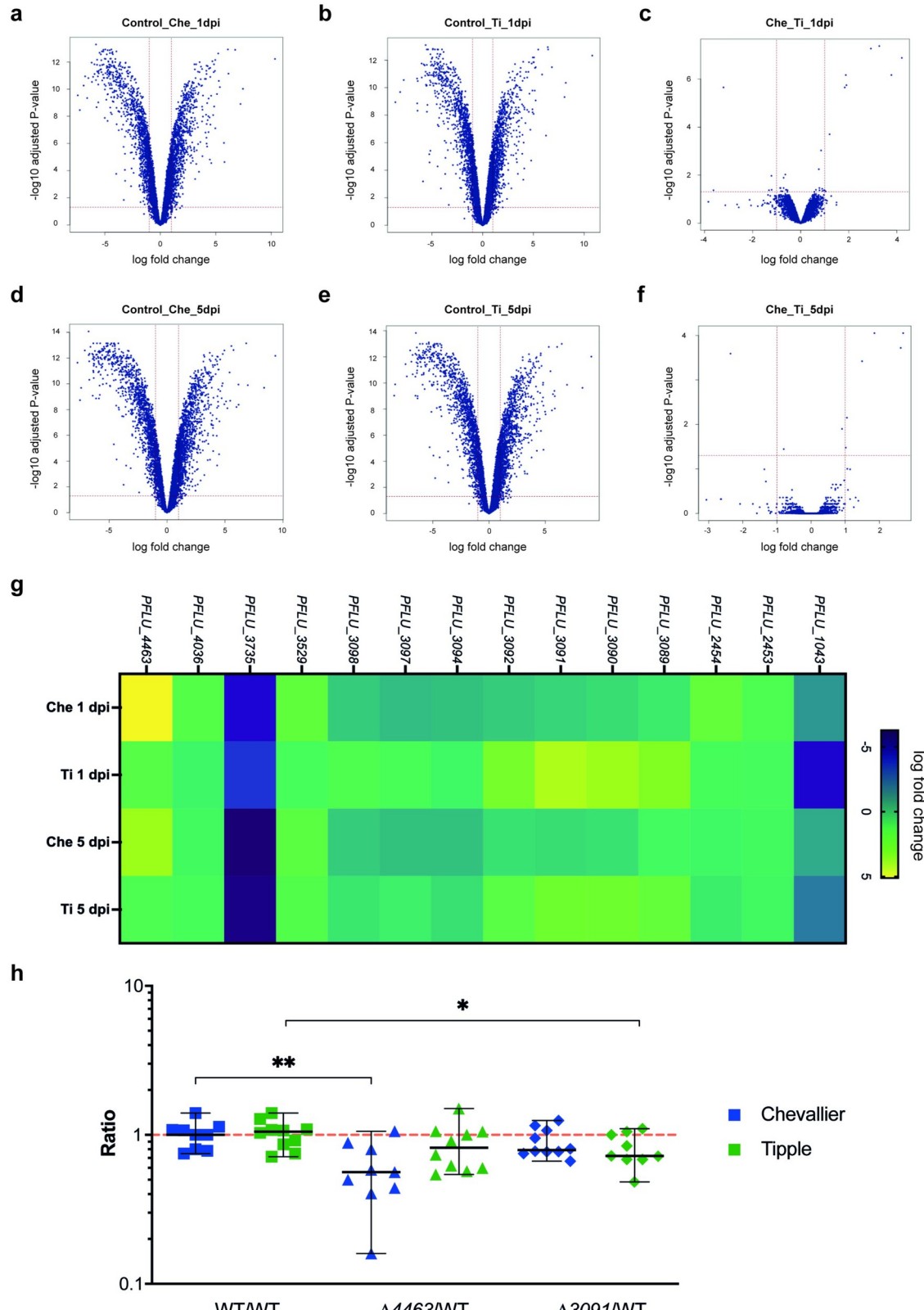

**Fig 6. SBW25 gene expression is affected by plant cultivar.** (A) Volcano plot comparison of SBW25 transcript abundance between liquid culture control and Chevallier rhizosphere at 1 dpi. (B) Volcano plot comparison of SBW25 transcript abundance between

liquid culture control and Tipple rhizosphere at 1 dpi. (C) Volcano plot represents the pairwise comparison of transcript abundance between SBW25 extracted from the Chevallier rhizosphere and from the Tipple rhizosphere at 1 dpi. (D–F) Volcano plots comparing transcript abundance at 5 dpi for the samples shown in A–C, respectively. (G) Heat map showing significantly differentially expressed genes in SBW25 between the Chevallier and Tipple rhizosphere samples at 1 dpi and 5 dpi (>1 log2-fold change from liquid culture control, $p$-values < 0.05). (H) Rhizosphere colonisation competition assay. The graph shows the ratios of Δ3091 and Δ4463 to SBW25-lacZ. CFUs recovered from the rhizospheres of Chevallier (blue) and Tipple (green) barley cultivars. Each dot represents the ratio of CFUs recovered from an individual plant; 8–10 plants were used per condition and $p$-values were calculated by Mann–Whitney U test, asterisks indicate $p < 0.05$ (*) and 0.01 (**). Experiment was repeated twice, and a representative graph is shown here. The data underlying this figure can be found in S6 Data. dpi, days post-inoculation.

microbiome. Furthermore, *Pseudomonas* SynComs promote Chevallier, but not Tipple plant growth.

## Discussion

Modern cereal varieties, such as Tipple, have been intensively bred for positive agricultural traits including high yield and seed starch content, short straw, and good malting properties. However, while incorporation of these positive traits has substantially improved yield and crop quality, the wider effect of these changes on plant physiology and ecological impact, and hence the sustainability of the resulting crops, are poorly understood. Consequently, there has been considerable recent effort to unravel the effect of plant genotype on the composition of the associated microbiota [6,27,30,33,46,47].

While several studies have demonstrated the importance of plant genes [29] or metabolic pathways [32] for microbial recruitment, resolution within the microbiome itself has typically been restricted to genus (or occasionally species) level changes for whole microbiome studies. Studies using synthetic microbial communities [6,45] enable interactions between microbes and hosts to be studied in greater detail; however, these communities typically only contain a few representatives of each genus, limiting their utility to probe the importance of bacterial genotype in host colonisation. This issue also cannot be fully addressed by molecular analyses of bacterial colonisation pathways [43,46], where the importance of individual loci in the highly complex plant environment is often difficult to extrapolate from laboratory results. To address these limitations, we combined amplicon metabarcoding and microbial isolation from

**Table 2. Predicted functions of genes differentially expressed in SBW25 between the Chevallier and Tipple rhizospheres.** The chromosomal organisation of genes *PFLU_3089* to *PFLU_3098* is represented in S5 Fig.

| Gene ID | Predicted function of encoded protein | Up-regulated in: |
|---|---|---|
| *PFLU_3089* | Hypothetical protein with a porin-like domain | Tipple |
| *PFLU_3090* | NAD-dependent aldehyde dehydrogenase | Tipple |
| *PFLU_3091* | Amino acid permease | Tipple |
| *PFLU_3092* | Methyl-accepting chemotaxis protein | Tipple |
| *PFLU_3094* | Amino acid permease | Tipple |
| *PFLU_3096* | 4-aminobutyrate aminotransferase | Tipple |
| *PFLU_3097* | NAD-dependent aldehyde dehydrogenase | Tipple |
| *PFLU_3098* | Cytochrome b-561 membrane protein | Tipple |
| *PFLU_3735* | Hypothetical protein | Tipple |
| *PFLU_1043* | Hypothetical protein | Chevallier |
| *PFLU_2454* | Enoyl-CoA hydratase | Chevallier |
| *PFLU_3529* | Acyl-CoA dehydrogenase | Chevallier |
| *PFLU_4036* | 4-aminobutyrate aminotransferase | Chevallier |
| *PFLU_4463* | EamA-like transporter family | Chevallier |

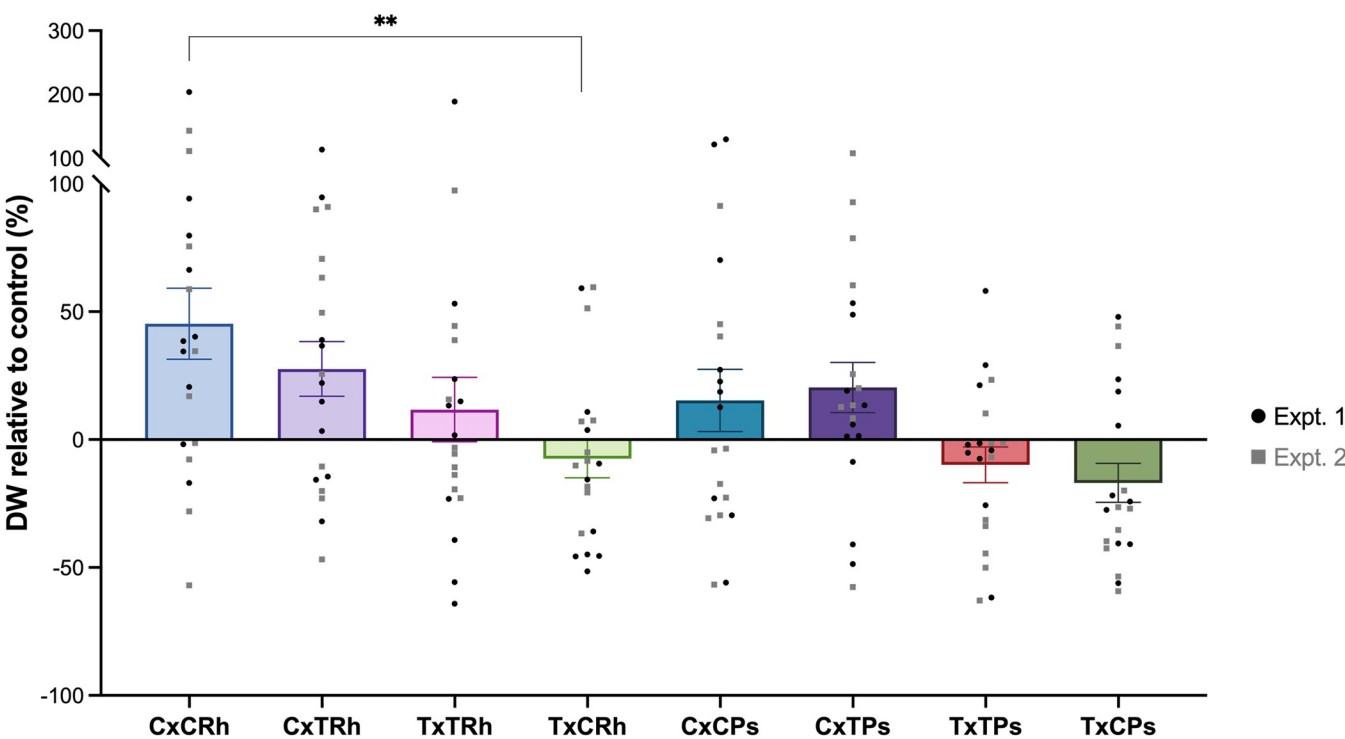

**Fig 7. Cross-inoculation assay between Chevallier and Tipple.** Chevallier (C) or Tipple (T) seedlings were inoculated under controlled conditions either with a rhizosphere extract (CRh/TRh) or with a SynCom of rhizosphere *Pseudomonas* (CPs/TPs). The graph summarises the dry weights (DW) of inoculated plants at 3 weeks post inoculation relative to the uninoculated controls. Two biologically independent repetitions of the experiment are represented side by side and up to 10 plants were used per condition. Data are presented as mean +/− std error; *p*-values were calculated by Tukey's multiple comparison test and asterisks indicate $p < 0.05$ (*), 0.01(**). The data underlying this figure can be found in S7 Data.

the rhizospheres of greenhouse-grown barley plants with genomic analysis and molecular microbiology to examine the molecular mechanisms driving cultivar-specific barley colonisation by the rhizobacterial genus *Pseudomonas* [2].

Individual plant species consistently recruit a similar core microbiome that persists regardless of the soil properties, cultivar and even isolation continent [47]. Consistent with this, we observed a substantial degree of overlap between the plant-associated bacterial microbiome samples, with significant enrichment of multiple plant-associated genera, including *Pseudomonas*, compared to bulk soil. Interestingly, the rhizospheres and root-associated microbiota of Chevallier and Tipple barley cultivars also exhibited striking differences, supporting a degree of cultivar-specific microbiome assembly. One such difference was the apparent inability of Tipple to influence the fungal composition of the rhizosphere, in marked contrast to Chevallier, where *Candida* species were almost excluded while *Saitozyma* species were strongly enriched. Focussing on *Pseudomonas*, Chevallier samples displayed a lower abundance of these bacteria than Tipple but higher overall microbial diversity, consistent with the increased complexity of root exudates seen for this landrace.

The differences in *Pseudomonas* recruitment between Chevallier and Tipple were also observed at the level of genotype, with isolated rhizosphere *Pseudomonas* spp. clustering according to their origin. Despite clear evidence for cultivar-dependent genotypic clustering, we observed little overlap between the 2 biologically independent experiments shown in Fig 2. This is consistent with the underlying soil microbiota exerting a dominant impact on the resulting rhizosphere microbiome [20], but with the barley plants exerting cultivar-specific

discrimination on the soil microbiome they find themselves growing in. Bulk soil isolates were evenly distributed throughout the phylogenetic tree, as expected given the soil is the source for most subsequent rhizosphere microbial recruitment.

Our data support a leading role for root exudates in cultivar-specific rhizosphere microbiome assembly in barley. The importance of exudates for root microbe recruitment has been widely studied [32,48,49]. Exudates are composed of sugars, amino acids, carboxylic acids, and phenolic compounds, alongside secondary metabolites such as hormones [41], with exudation profiles varying with plant developmental stage, age, plant species, and genotype [50]. In barley exudates, the previously uncharacterised flavonoid saponarin has been shown to exert allelopathic properties against weeds such as *Bromus diandrus* [51]. However, comparatively little is known about the relationship between barley root exudates and microbiome assembly.

GC-MS analysis identified several significant differences in the abundance of numerous metabolites between the Chevallier and Tipple exudates, with Tipple exudates containing a less diverse array of compounds overall than Chevallier. We hypothesised that differences in the exudation of key molecules might impact the subsequent establishment of specific microbial communities. The reduced molecular diversity and higher prevalence of hexose sugars, such as glucose and fructose seen in Tipple exudates might therefore explain the greater abundance, but lower phenotypic and genetic diversity observed for the Tipple-associated *Pseudomonas* isolates. In support of this, a *P. fluorescens* mutant (Δ*rccR*) that cannot regulate the glyoxylate shunt and struggles to grow on hexose sugars [43] was significantly compromised in the Tipple rhizosphere, but much less so in Chevallier. The capacity to utilise available carbon resources is clearly a key factor in bacterial adaptation to the rhizosphere environment and is under selection in our experiments. *Pseudomonas* strains that grew well on fructose/glucose were more abundant among the Tipple rhizosphere isolates than isolates from Chevallier.

Analysis of the genomic features present in plant-associated bacterial populations is a powerful tool to identify genes that play important roles in adaptation to plant environments [24,52]. By examining the frequency of previously identified plant-association loci [44] in the genomes of sequenced barley isolates, we identified a series of genes whose abundance differed significantly between Chevallier and Tipple-derived isolates. A subset of these loci was then deleted in *P. fluorescens* SBW25 and their contribution to in planta fitness compared for the 2 cultivars. Using a single, well-characterised model organism for these assays enabled us to directly compare the impact of the tested genes with one another. Using this approach, we identified 3 new barley cultivar-discrimination genes: *PFLU2583*, a sugar transporter, *PFLU6072*, a transcription factor predicted to play a role in carbon metabolism control, and *PFLU5080*, an uncharacterised protein with a putative role in prophage excision. Differential selection of *PFLU6072* and *PFLU2583* aligns well with the hypothesis that carbon metabolic adaptation is central to successful plant colonisation [43,53]. The importance of *PFLU5080* is less obvious, although phage-mediated regulation of host behaviour has been identified in several microbes, such as *E. coli* where phage-infected bacteria were shown to outcompete non-infected bacteria under limiting carbon conditions [54]. *PFLU5080* may play an analogous role here, modulating bacterial host responses to environmental differences.

While we identified several cultivar-specific *Pseudomonas* colonisation genes, 5 of the 8 mutants we tested showed little/no fitness defect compared to WT SBW25. This could be due to several reasons, such as differences between our gnotobiotic experimental setup and the original selection environment, genetic differences between SBW25 and the genotypes where these loci were selected, or a degree of functional redundancy in SBW25. This was not entirely unexpected: in a similar study of regulatory genes with potential roles in SBW25 wheat colonisation, only 2 of 7 mutants displayed clear fitness defects, despite all 7 showing altered

colonisation phenotypes and increased expression in the wheat rhizosphere, consistent with bona fide roles in rhizosphere colonisation [53,55].

RNA-seq analysis of SBW25 rhizosphere gene expression identified additional loci whose importance differed between Tipple and Chevallier. In addition to around 1,000 plant-induced genes whose mRNA abundance increased in both rhizospheres, we identified and tested a subset of cultivar-specific genes. Interestingly, only 14 loci were differently expressed between the 2 cultivars, with 9 up-regulated in the Tipple rhizosphere and 5 in Chevallier. *PFLU3091*, an amino acid permease up-regulated in the Tipple rhizosphere and *PFLU4463*, a drug/metabolite transporter up-regulated in Chevallier were subsequently shown to be differentially important for colonisation, in agreement with the expression data. These findings demonstrate that barley genotype exerts selective pressure on the *Pseudomonas* population at every organisational level, affecting overall abundance, the frequency of genotypes and individual genetic features and the expression of specific loci within individual microbes.

Previous studies have shown that plants secrete specialised secondary metabolites, such as coumarins [31] or triterpenes [32] to influence their microbiome composition. Our results suggest that higher-level mechanisms for microbial selection also exist, where broad differences in the secretion of primary metabolic compounds can shape the microbiome towards the interests of the plant. The extent to which these 2 processes for microbiome recruitment coexist is currently unclear, although it seems plausible that the specialised metabolic shaping described by the Pieterse, Osbourn and Bai labs [31,32] might act to refine the more general influence of primary exudate secretion we describe here.

While the plant-beneficial properties of rhizosphere microbes are well known [13], the reasons why plant microbiomes differ between members of the same species are less clear. Why do different barley varieties, growing in the same soil environment, recruit different populations of beneficial rhizobacteria and fungi? Excitingly, our results support a cultivar-dependent link between plant growth and the composition of the recruited microbiome, suggesting that the microbiome shaping we see here has real consequences for plant health: plants recruit specific microbes because these microbes help them to grow. Our data suggest that the degree of population fine-tuning that takes place and the consequences of this for plant fitness are both broader and more complex than we previously suspected.

A major challenge for reductionist analyses of the plant microbiome is in translating the findings of greenhouse studies to real-life scenarios. The core phenomenon that we describe here, of root exudate-driven shaping of the rhizosphere microbiome, appears to be robustly maintained across different laboratory environments, from axenically grown seedlings to plants growing in potting compost with a complex (albeit not agriculturally representative) microbiome. Nonetheless, it is entirely possible that phenomena identified under controlled lab/greenhouse settings may not translate predictably to complex, uncontrolled field environments. Determining the extent to which the phenomena we describe here can be observed for crops growing in farm fields is a key challenge for future research.

## Materials and methods

### Biological material and growth conditions

Barley cultivars Chevallier and Tipple were used throughout this study. All seeds were surface sterilised prior to use with 70% ethanol for 1 min, 5% sodium hypochlorite for 2 min followed by thorough washing with sterile distilled water. Following sterilisation, seeds were germinated on 1.5% water agar plates in darkness and at room temperature (RT) for 48 h or until germination before further use.

Bacterial strains used in this work are listed in S5 Table, with plasmids listed in S6 Table and primers in S7 Table. *Pseudomonas* strains were grown overnight at 28˚C with shaking in lysogeny broth (LB) [56], King's medium B (KB) [57], or M9 minimal media supplemented with carbon sources at a final concentration of 0.4% w/v [56], as stated in the text. *Pseudomonas* growth media was supplemented with antibiotics and other additives as described elsewhere in the text. *Rhizobium leguminosarum* biosensor strains [38] were grown overnight at 28˚C with shaking in tryptone yeast (TY) [58] or universal minimal salts (UMS) [38] medium. UMS was supplemented with 30 mM pyruvate and 10 mM ammonium chloride unless specified otherwise and solidified with 1.5% agar (UMA) where appropriate. Antibiotics were added, when necessary, at the following concentrations: streptomycin (500 μg/ml) and tetracycline (2 μg/ml in UMS, 5 μg/ml in TY). *Escherichia coli* strains were grown in LB medium and plates solidified with 1.5% agar at 37˚C. Media was supplemented with 12.5 μg/ml tetracycline where necessary. *Streptomyces* venezuela*e* ATCC 10712 was grown at 28˚C until sporulation on MYM medium supplemented with 2% agar [59].

## Isolation of root-associated *Pseudomonas*

Following germination, barley seedlings were transplanted into 9 cm pots containing JIC (John Innes Centre) Cereal Mix (40% Medium Grade Peat, 40% Sterilised Soil, 20% Horticultural Grit, 1.3 kg/m$^3$ PG Mix 14-16-18 + TE (trace elements) Base Fertiliser, 1 kg/m$^3$ Osmocote Mini 16-8-11 2 mg + TE 0.02% B, Wetting Agent, 3 kg/m$^3$ Maglime, 300 g/m$^3$ Exemptor). Plants were grown for 3 weeks in a controlled environment room (CER) at 25˚C with a 16 h light cycle and watered twice a week with sterile tap water. The rhizospheres of 3 independent plants from each cultivar were sampled individually. Flame-sterilised scissors were used to remove the shoots from each plant and excess soil was removed from the root system by energetic shaking before transferring to sterile 50 ml tubes, and 30 ml bulk soil control samples were taken from the centre of unplanted pots. Each tube was filled up to 50 ml with sterile phosphate-buffered saline (PBS) and vortexed for 10 min at 4˚C. Dilution series were produced in sterile PBS and plated onto *Pseudomonas* Agar Base, supplemented with CFC (cetrimide/fucidin/cephalosporin, Oxoid, United Kingdom) and prepared according to the manufacturer's instructions. Plates were incubated at 28˚C until visible colony formation. Single colonies were then randomly selected, streaked to single colonies on KB agar plates and cryopreserved for further analysis; 20 isolates were randomly selected per plant, for a total of 120 isolates per cultivar across 2 biologically independent experiments.

## Phenotypic and genotypic characterisation of isolated *Pseudomonas* spp

Barley rhizosphere *Pseudomonas* isolates were phenotyped for several ecologically relevant traits as follows. A high-throughput screening protocol was developed in which large agar plates were inoculated using a microplate replicator, enabling hundreds of isolates to be screened in parallel; 500 cm$^2$ (22.5 cm/side) square plates were used to assess Congo red binding, UV fluorescence, and protease activity (384 isolates tested/plate). Approximately 140 mm Petri dishes were used to study *S. venezuelae* suppression (96 isolates/plate) and motility and 96-multiwell plates were employed to test HCN production. Fluorescence emission under UV light (a proxy for siderophore production) was visually assessed after 48 h growth on KB agar [60]. To assess differences in polysaccharide and proteinaceous adhesin production, 0.005% w/v Congo red dye was added to KB agar and differences in colony pigmentation were assessed after 48 h [60]. Protease production was assessed as the ability to visually degrade 1% w/v milk powder added to KB agar plates after 48 h growth. Motility was studied by observing the spreading patterns of colonies grown for 24 h on 0.5% agar KB plates. Cyanogenic bacteria

were detected using an adaptation of the method of Castric and Castric [61], with cultures grown in 96-well plates, after [24]. *S. venezuelae* suppression was assessed for *Pseudomonas* overnight cultures spotted onto a lawn of *Streptomyces* spores (200 μl of a 1:25 suspension spread onto 140 mm Petri dishes containing Difco Nutrient Agar (DNA, Thermo Fisher, United States of America). Growth and inhibition of both microbes was assessed 10 dpi.

For each assay, ordinal values between 0 and 3 were assigned to each sample, except for protease activity and motility assays where 0 (phenotype absent) and 1 were assigned. Representative phenotypes for each ordinal value are shown in S8 Fig. Phenotyping assays were conducted at least twice independently. Where disagreements were recorded in the ordinal data, additional repeats were conducted until a firm consensus was reached.

The *gyrB* housekeeping gene was used to identify bacterial isolates, enabling resolution to the strain level. Briefly, colony PCR was performed using the primers described by Yamamoto and Harayama [35] and PCR products were sent for Sanger sequencing using both forward and reverse primers (Eurofins Genomics, Germany). Sequence alignments were performed in MEGA X 10.1 [62] and Geneious (Geneious Prime 2021.1.1) by MUSCLE [63]. Sequences were all trimmed at the maximum length of the shortest query, 816 bp, and subsequent phylogenetic analyses were carried out. MEGA X 10.1 [62] and iTOL [64] were used for the construction of the resulting tree, utilising the maximum-likelihood method and Tamura–Nei model with a bootstrap value of 1,000.

## Illumina whole genome sequencing and reciprocal BLAST analysis

The 22 Chevallier and 20 Tipple-derived *Pseudomonas* isolates were selected for whole genome sequencing according to 3 criteria: (i) phylogenetic distribution to ensure maximum diversity; (ii) growth efficiency in different carbon sources; and (iii) random selection to minimise bias. Genomic DNA (gDNA) was extracted from overnight LB liquid cultures using a GenElute Bacterial Genomic DNA Kit (Sigma-Aldrich, USA) following the manufacturer's instructions. gDNA quality was evaluated using a NanoDrop ND-1000 Spectrophotometer (Thermo Scientific, USA) and quantity measured with a Qubit 2.0 Fluorometer using high sensitivity buffer (Thermo Scientific, USA). Sample concentration was normalised to 30 ng/μl and 20 μl of each sample was sent for genome sequencing to the Earlham Institute (Norwich Research Park, United Kingdom). LITE libraries were prepared, and Illumina short read sequenced using NovaSeq6000 SP with 150 paired-end reads, aiming for 30× coverage [65].

Genome assembly was performed using SPAdes 3.13.1 and default settings [66], then nucleotide sequences were annotated for the presence of 410 genes of interest using BLAST and custom Perl scripts [67]. The library of genes of interest included known plant-induced genes: transporters, biofilm formation regulators, chemotaxis proteins, and siderophore pathways [44] as well as prominent substrate transporters and transcriptional regulators [13,43]. *P. fluorescens* SBW25 [44] was used as a reference genome. Following reciprocal BLAST analysis, genomes were manually examined for the presence or absence of individual genes. Decisions were made based on a combination of sequence identity (>66%), alignment coverage (>66%), and whether each match was a reciprocal best hit. Candidate genes for further mutational analyses were selected manually based on significance (Chi-squared test, $p < 0.05$). Genomic data has been deposited in ArrayExpress with accession number E-MTAB-12917.

## 16S/ITS amplicon sequencing

Five Tipple and 5 Chevallier barley plants were grown for 3 weeks in a CER at 25˚C with a 16 h light cycle and watered twice a week with sterile tap water, alongside bulk soil (cereal mix) controls. About 5 g of the rhizosphere-root sample was removed from each plant and decanted

into sterile 50 ml tubes. Samples were covered with 10 to 20 ml of 0.1 M $KH_2PO_4$ (pH 8.0) buffer and incubated for 30 min at RT with shaking. This first suspension was kept, and roots were transferred into fresh 50 ml tubes. Another 10 to 20 ml of buffer was added, and this washing process of the roots was repeated a total of 3 times, every time keeping the supernatant separately (wash 1, wash 2, and wash 3). For the last step, 10 to 20 ml were added, and tubes were vortexed twice for 30 s, roots were removed and placed in a new tube and this last wash (wash 4) was collected. Wash 1 and wash 2 were pooled and centrifuged for 10 min at $29,000 \times g$. Supernatant was discharged, wash 3 and 4 were combined into this mix and a final centrifugation was carried out. The resulting pellet was considered to represent the total rhizosphere microbiome. Washed roots were assumed to contain only closely associated epiphytes and endophytes [68].

gDNA extraction was performed using a FastDNA SPIN Kit for soil (MP Biomedicals, UK) according to the manufacturer's instructions. gDNA quality was evaluated using the Nano-Drop ND-1000 Spectrophotometer (Thermo Scientific, USA) and quantity measured with a Qubit 2.0 Fluorometer using high sensitivity buffer (Thermo Scientific, USA). Following gDNA extraction, a quality control step was conducted with the same input material in all cases. The samples were sent for library preparation and sequencing to Novogene Co. (Hong Kong, CN) using Illumina NovaSeq6000 with 250 paired-ends, aiming for 50,000 reads/library. The regions targeted were V4 for the bacterial 16S rRNA gene and ITS1-1F for fungal ITS.

Amplicon sequencing data analysis was conducted on the demultiplexed files as described previously [68] with some modifications. As no libraries were observed to fail or deliver low read numbers, we refrained from rarefying. Briefly, on the demultiplexed files supplied with primers and sequence adapters already removed, fastp version 0.20.0 [69] was run on 16S as well as ITS data with disabled length filter and trim_poly_g to remove polyG read tails. Following this, data quality was controlled by R-3.6.3 and DADA2 version 1.14.1 according to the workflow version 2 described in [70]. The truncation length for forward reads was set to 200 bp and for reverse reads to 180 bp for 16S and 210 bp for ITS, respectively. For 16S and ITS libraries, the following parameters were used: maxN = 0, maxEE = c(2, 2), and truncQ = 2 and a minimum length of 50 bp. In both cases, forward and reverse reads were merged with default settings. Silva database (silva_nr_v132) was used to classify bacterial reads [71] and UNITE (sh_general_release_dynamic_s_01.12.2017) for fungal reads [72]. Reads without a match in the databases used were removed. Alpha-diversity analysis was based on Shannon index and Observed measure, calculated on pre-normalised data (package "phyloseq," R-3.6.3, version 1.30.0). The statistical analysis for the alpha-diversity was decided as follows: First, a Shapiro test was conducted to establish if the data was normally distributed. If the data followed a normal distribution, a one-way ANOVA with type I sums of squares was used. If the ANOVA result was significant, it was followed up by a Tukey HSD test for pairwise comparison between groups. In case of non-normal distributed data, a Kruskal–Wallis test, as the nonparametric equivalent of the ANOVA, was performed. If the result was significant, this was followed by a Wilcoxon Rank Sum test for pairwise comparison. The following pairings were tested: differences between the 2 cultivars and the bulk soil, differences of niches between cultivars, and differences of niches within each cultivar. Data from both cultivars was pooled in each case. Regarding beta-diversity analysis, ASVs with a mean lower than $10^{-5}$ were ignored for subsequent analysis, and the filtered ASV data was used to calculate Bray–Curtis beta-diversity (R-3.6.3 "vegan" package, version 2.5.6). Statistical analyses were also performed on filtered data by using the package "vegan", ANOSIM and PERMANOVA: adonis function [73]. For beta-diversity statistical testing either ANOSIM (analysis of similarity) or PERMANOVA (permutation analysis of variance) were used. For PERMANOVA, the assumption is that the beta dispersion between groups is equal. This was tested using the betadisper test. If the beta

dispersion between groups was not equal, we used ANOSIM as our statistical test. For data visualisation, ggplot2 (version 3.3.0) was used.

## Bacterial genetic manipulation

Vectors and primers used in this work are listed in S4 and S5 Tables. Gene deletions in SBW25 were made by allelic exchange following a two-step homologous recombination process [74]. To summarise, 500 to 700 bp homologous flanking regions of the target regions were either PCR-amplified and Gibson-assembled into the BamHI site of the suicide vector pTS1 [75] or synthetically designed by Twist Biosciences (California, USA). The pTS1 vector is an adaption of pME3087 [76] containing a *sacB* gene which allows counter-selection on sucrose plates.

*P. fluorescens* SBW25 electrocompetent cells were prepared by growing the cells overnight and washing with 300 mM sucrose 3 times at RT. These cells were then electroporated at 2,500 V with 100 to 300 ng of the gene deletion constructs. Cells were recovered in 3 ml of LB and incubated for 2 h at 28˚C with shaking to enable expression of antibiotic resistance genes. Single crossovers were selected on 12.5 μg/ml tetracycline and re-streaked to obtain individual colonies. These were then grown overnight without selection in 50 ml of LB at 28˚C with shaking. Dilution series of each culture were plated on LB containing 10% sucrose to counter-select bacteria without a second homologous recombination event. Individual sucrose resistant colonies were patched on LB +/− tetracycline to confirm double recombinants and successful gene deletions confirmed by PCR.

Deletion of the *rccR* gene in the SBW25-*lacZ* background was conducted using the pME3087 construct for *rccR* deletion described in [43]. This deletion vector was transformed by electroporation into SBW25-*lacZ*, and single crossovers selected on 12.5 μg/ml tetracycline and re-streaked to single colonies. Cultures from single crossovers were grown overnight in LB, then diluted 1:100 into fresh medium. After 2 h, 5 μg/ml tetracycline was added to inhibit the growth of cells that had lost the tetracycline cassette. After a further hour of growth, samples were pelleted and re-suspended in fresh LB containing 5 μg/ml tetracycline and 2 mg/ml piperacillin and phosphomycin to kill growing bacteria. Cultures were grown for a further 4 to 6 h, washed once with LB, and dilution series were plated onto LB agar. Resulting colonies were patched onto LB +/− tetracycline, and Tet-sensitive colonies tested for gene deletion by colony PCR.

## Bacterial growth assays

Assays were conducted in 96-multiwell plates containing 199 μl of M9 medium supplemented with carbon sources as described in the main text, and 1 μl of bacterial overnight LB cultures were inoculated into each well using a multichannel pipette, providing an initial $OD_{600}$ of approx. 0.01. Plates were incubated statically at 28˚C for 48 h and bacterial growth was monitored using a microplate spectrophotometer, either SPECTROstar Nano (BMG LABTECH, UK) or PowerWave (BioTek Instruments, USA). Each experiment was independently repeated at least twice.

## Root colonisation assays

Germinated seedlings were placed into sterile 50 ml plastic tubes containing washed, medium grain vermiculite and rooting solution (1 mM $CaCl_2 \cdot 2H_2O$, 100 μm KCl, 800 μm $MgSO_4$, 10 μm FeEDTA, 35 μm $H_3BO_3$, 9 μm $MnC_{12} \cdot 4H_2O$, 0.8 μm $ZnCl_2$, 0.5 μm $Na_2MoO_4 \cdot 2H_2O$, 0.3 μm $CuSO_4 \cdot 5H_2O$, 6 mM $KNO_3$, 18.4 mM $KH_2PO_4$, and 20 mM $Na_2HPO_4$). Seedlings were grown for 7 days at 25˚C with 16 h light cycle, then inoculated with a 1:1 mix of $1 \times 10^3$ CFUs WT and mutant SBW25 strains. Plants were grown for a further 5 days, after which shoots

were removed and 20 ml of PBS was added to each tube and vortexed for 10 min at 4°C to resuspend bacteria. Dilution series were then plated onto LB supplemented with 100 μg/ml carbenicillin, 0.1 mM IPTG, and 50 μg/ml X-Gal. Plates were incubated at 28°C until blue and white colonies were clearly distinguishable. Colony counting was undertaken, and final blue/white ratios were calculated for all the mutants tested. Eight to 10 plants were sampled per condition and experiments were repeated at least twice independently.

## Root exudate screening with Rlv3841_lux biosensors

The method described in [38] was adapted for optimal results in barley plants. Square 100 mm plates were filled in angle with 75 ml Fahraeus agar (FP) [77], a small square section was pierced on top of each plate to allow growth of the barley seedling and the medium was covered with sterile filter paper upon which 1 seedling was placed. Biosensors were grown on UMS 1.5% agar slopes for 3 days at 28°C, then washed 3 times in UMS medium without any additions, and 200 to 300 μl of each Rlv3841_lux biosensor [38] suspension adjusted to an $OD_{600}$ of 0.1 was inoculated directly onto the seedling roots. A second filter paper was applied on top of the inoculated seedling and aluminium foil-wrapped plates were placed in a CER at 25°C with 16 h light cycle until photographed. A NightOWL II LB 983 CCD camera-box (Berthold Technologies GmbH & Co., Germany) was used to take pictures of plates after 2, 5, and 7 dpi. ImageJ software [78] was used to analyse the resulting pictures. The total luminescent area present in each plate was evaluated and relative values were recorded and compared.

## Root exudate extraction and GC-MS analysis

To examine bacterial growth on root exudates, 4 sterile barley seedlings/50 ml tube were grown hydroponically in 40 ml of sterile rooting solution, in a CER at 25°C with 16 h light cycle. After 3 weeks, 80 ml of solution per plate were concentrated using a Genevac, resuspended in water, filter-sterilised, and used to supplement 50 ml water agar plates; 3 μl spots of bacterial overnight cultures were inoculated onto these plates and incubated for 3 days at 28°C, until colonies emerged.

For GC/MS analysis, a sterile hydroponic system was constructed and used to grow barley plants for up to 3 weeks [30]. Two 50 ml plastic tubes were connected with muslin fabric and the bottom part was filled with rooting solution. The whole system was autoclaved to ensure sterility. Barley seedlings were transferred into the tubes and grown in a CER at 25°C with 16 h light cycle for 3 weeks. The system was topped up with filter-sterilised fresh rooting solution once a week. The root exudates extraction was performed as described elsewhere [30]. Plants were removed from the tubes and roots were carefully washed with sterile deionised water. Four plants were transferred into 200 ml of milli-Q water and incubated for 2 h in the same growing conditions. The liquid fractions were then freeze-dried and diluted with 80% methanol to a concentration of 10 μg/μl. Organic solvent was removed by placing samples on a GeneVac (SP Scientific) prior to processing.

Samples were analysed using Agilent GC-MS Single Quad (7890/5977) plus Gerstel Multi-Purpose Sampler (MPS, Agilent Technologies, USA) following the Agilent G1676AA Fiehn GC/MS metabolomics workflow. The MPS enables the automated derivatisation of samples and their subsequent processing. Identification of molecules was based on comparison with the metabolic libraries Agilent Fiehn 2013 GC-MS Metabolomics RTL (Agilent Technologies, USA) and the NIST17 Version 2.3 GC Method Retention Index Library. The Agilent MassHunter Workstation package, particularly Qualitative, Unknowns and Mass Profiler Professional software were used for identification, analysis, and data visualisation (Agilent Technologies, USA).

## Rhizosphere RNA extraction and RNA-seq analysis

SBW25 RNA was extracted from the rhizospheres of Chevallier and Tipple growing in axenic conditions as described above for the colonisation assays. The 7-day-old seedlings were inoculated with 1 ml of a PBS-washed *P. fluorescens* SBW25 culture adjusted to $OD_{600}$ = 1.0 and RNA was extracted at 1 and 5 dpi, and 5 ml samples of density-adjusted cell culture prior to plant inoculation were pelleted for 10 min at 9,000 × g at 4˚C and included as a control. Pellets were flash-frozen and kept for later RNA extraction.

For the rhizosphere samples, 12 plants were combined in each sample to provide sufficient RNA. Three biologically independent rhizosphere samples were extracted per time point. Aerial parts of the plants were removed with sterile scissors and 8 ml of PBS and 12 ml of RNA later were added to each plant tube. The sample tubes containing vermiculite and separated root systems were then vortexed for 10 min at 4˚C to resuspend bacteria and combined to create a single rhizosphere sample. Immediately, samples were filtered through 4 layers of previously autoclaved muslin cloth placed in a sterile glass funnel and collected into sterile centrifuge bottles. The filtrate was centrifuged at 170 × g at 4˚C, the supernatant was transferred to a new bottle and the centrifugation step was repeated to remove any residual vermiculite. The supernatant was transferred to a new bottle and centrifuged for 10 min 17,000 × g and 4˚C. Cell pellets were then flash-frozen for further use. To lyse the cells, pellets were resuspended in 400 μl of 10 mM Tris-Cl (pH 8.0) and 700 μl of RLT buffer with β-mercaptoethanol and transferred to Matrix B tubes (MP Biomedicals, UK). Samples were lysed with two 90-s pulses in a FastPrep machine (MP Biomedicals, UK) with 90-s rest on ice between them. Samples were centrifuged for 15 min at 16,000 × g and 4˚C, and RNA extracted from the supernatant with an RNeasy Mini kit (Qiagen, GE). On-column digestion was conducted with an RNAse-Free DNase Set (Qiagen, GE). RNA quality and concentration was checked by a nanodrop spectrophotometer and a Qubit 2.0 Fluorometer with dilution steps conducted where necessary. Samples were subjected to a second DNase treatment with a TURBO DNA-free Kit (Thermo Scientific, USA) and RNA integrity was confirmed on a 1% agarose gel.

Samples were sent to Novogene Co. (Hong Kong, CN) for rRNA depletion, strand-specific library construction and sequencing on an Illumina NovaSeq 6000. Subread [79] was used to align the reads in the fastq files to the SBW25 reference genome. BAM files were sorted and indexed using SAMtools (version 1.8) [80]. Mapping of the reads were counted from the BAM files using the featureCounts program, part of the Subread package, resulting in a table of counts with one row for each gene and one column for each sample. The table of counts was used as the input to the R package edgeR [81] to test differential expression of each gene and a differential log fold changes table was produced. RNA-seq data has been deposited in ArrayExpress, with accession number E-MTAB-12918.

## Cross-inoculation experiment

The rhizospheres of 3-week-old barley plants grown in Cereal mix as described above were extracted by pulling the plants from the pots and firmly shaking the roots. The rhizosphere inoculum of each barley cultivar consisted of the combined sample from 4 plants. Shoots were removed and up to 5 g of root material were placed in 50 ml Falcon tubes, which were filled with PBS buffer and vortexed for 10 min at 4˚C to resuspend bacteria. Roots were discarded and the remaining PBS was transferred to a larger container. Heavier particles were removed, and the rhizosphere suspension was centrifuged at 2,700 × g for 30 min at 4˚C, washed with PBS, and the process was repeated 3 times. The final pellet of each cultivar was resuspended into 200 ml so that a sufficient volume of inoculum for each plant could be produced. Each seedling was inoculated with 5 ml of this rhizosphere suspension.

The *Pseudomonas* synthetic communities (SynComs) used in the cross-inoculation experiment were created by mixing the 60 strains originally isolated from Chevallier and Tipple roots (see Fig 2) in an equal ratio as described elsewhere [45] but with some modifications. Briefly, overnight cultures were adjusted using PBS instead of $MgCl_2$ to an $OD_{600}$ of 0.2, then mixed and re-adjusted to a final $OD_{600} = 0.2$ with PBS. Seedlings were inoculated with 5 ml of each SynCom inoculum.

The whole experiment was performed under controlled conditions at 25˚C with 16 h light cycle in hydroponic systems using vermiculite and rooting solution as described above. Plants were placed in trays separated by treatment and watering took place from the bottom every 2 or 3 days, 200 ml during the first week and 500 ml for the rest of the experiment. After 3 weeks, vermiculite was removed from the roots, plants were placed in an oven at 60˚C for up to 6 days and dry weight was recorded.

## Supporting information

**S1 Table. Species designations of barley root isolates based on the *gyrB* housekeeping gene.**
(XLSX)

**S2 Table. GC-MS detected compounds in root exudates of Tipple and Chevallier.**
(DOCX)

**S3 Table. Differentially present genes in the *Pseudomonas* populations and their functions.**
(DOCX)

**S4 Table. Gene expression data for SBW25 rhizosphere RNA-seq assays.**
(XLSX)

**S5 Table. Bacterial strains used in this work.**
(DOCX)

**S6 Table. Plasmids used in this study.**
(DOCX)

**S7 Table. Oligonucleotides used in this work.**
(DOCX)

**S1 Fig. Alpha diversity comparison for bacterial communities.** Observed richness and Shannon diversity were used as diversity measures. (A) Overall comparison between the rhizosphere communities of Chevallier and Tipple and the bulk soil. (B) Root endosphere community comparison between Chevallier and Tipple. (C) Comparison of community composition between Chevallier compartments. (D) Comparison of community composition between Tipple compartments. Five replicates, represented as different coloured dots, were used per condition. Asterisks indicate $p < 0.05$ (*), 0.01 (**), or 0.001(***). The data underlying this figure can be found in S8 Data.
(TIF)

**S2 Fig. Alpha diversity comparison for fungal communities.** Observed richness and Shannon diversity were used as diversity measures. (A) Overall comparison between the rhizosphere communities of Chevallier and Tipple and the bulk soil. Significant differences for both indices, between cultivars and Chevallier and bulk soil. (B) Root endosphere community comparison between Chevallier and Tipple. No significant differences were found. (C) Comparison of community composition between Chevallier compartments. (D) Comparison of community composition between Tipple compartments. Five replicates, represented as different coloured dots, were used per condition. Asterisks indicate $p < 0.05$ (*), 0.01 (**), or 0.001

(***). The data underlying this figure can be found in S9 Data.
(TIF)

**S3 Fig. Phenotypic traits evaluated in the *Pseudomonas* rhizosphere strains and their significance according to Chi-Square comparisons.** (A) Congo red binding (CRB). (B) Protease activity (PA). (C) Motility (MO). (D) CRB Chi-Square paired comparisons. (E) PA Chi-Square paired comparisons. (F) MO Chi-Square paired comparisons. (G) Fluorescence emission (FE). (H) Hydrogen cyanide production (HCN). (I) *Streptomyces* suppression (SS). (J) FE Chi-Square paired comparisons. (K) HCN Chi-Square paired comparisons. (L) SS Chi-Square paired comparisons. Data is shown as the relative percentage of isolates presenting a given score. Significant differences according to Chi-square test are represented. The data underlying this figure can be found in S10 Data.
(TIF)

**S4 Fig. Luminescent *Rhizobium* biosensors growing on UMA plates containing the indicated carbon sources at 10 mM concentration, apart from pyruvate (30 mM), succinate (20 mM), and phenylalanine (5 mM).** Aspartate, succinate, and phenylalanine plates were additionally supplemented with 30 mM pyruvate to enable bacterial growth. Biosensors used are listed in S5 Table.
(TIF)

**S5 Fig. Growth of the 42 sequenced *Pseudomonas* barley rhizosphere isolates after 3 days on water agar plates, containing Chevallier or Tipple root exudates as the sole nutrient source.**
(TIFF)

**S6 Fig. Growth curves of deletion mutants in SBW25 of genes differentially distributed between Chevallier and Tipple rhizosphere isolates at 36 h.** (A) LB medium. (B) KB medium. (C) M9 minimal medium 0.4% acetate. (D) M9 minimal medium 0.4% succinate. (E) M9 minimal medium 0.4% pyruvate. (F) M9 minimal medium 0.4% glucose. (G) M9 minimal medium 0.4% Glycerol. Three biological reps used per strain. Error bars are represented as SEM. Experiment was repeated 3 times and here a representative graph is shown. The data underlying this figure can be found in S11 Data.
(TIFF)

**S7 Fig. Genomic context of the *PFLU_3089–PFLU_3098* region of the SBW25 genome.** Gene numbers are indicated above each gene. Genes up-regulated in the Tipple rhizosphere relative to Chevallier are highlighted in green. Predicted encoded protein functions are given for up-regulated genes. Arrows indicate the direction of open reading frame transcription in each case.
(TIF)

**S8 Fig. Ordinal scales used to score bacterial phenotypes: Congo red binding (CRB), fluorescence emission (FE), hydrogen cyanide production (HCN), streptomyces suppression (SS), protease activity (PA), and motility (MO).**
(TIF)

**S1 Data. Dataset underlying Fig 1.**
(XLSX)

**S2 Data. Dataset underlying Fig 2.**
(TXT)

**S3 Data. Dataset underlying Fig 3.**
(XLSX)

**S4 Data. Dataset underlying Fig 4.**
(XLSX)

**S5 Data. Dataset underlying Fig 5.**
(XLSX)

**S6 Data. Dataset underlying Fig 6.**
(XLSX)

**S7 Data. Dataset underlying Fig 7.**
(XLSX)

**S8 Data. Dataset underlying S1 Fig.**
(XLSX)

**S9 Data. Dataset underlying S2 Fig.**
(XLSX)

**S10 Data. Dataset underlying S3 Fig.**
(XLSX)

**S11 Data. Dataset underlying S6 Fig.**
(XLSX)

## Author Contributions

**Conceptualization:** Alba Pacheco-Moreno, Chris Ridout, Sarah DeVos, Jacob G. Malone.

**Data curation:** Alba Pacheco-Moreno, Anita Bollmann-Giolai, Govind Chandra.

**Formal analysis:** Govind Chandra, James K. M. Brown, Paul Nicholson.

**Funding acquisition:** Chris Ridout, Sarah DeVos, Jacob G. Malone.

**Investigation:** Alba Pacheco-Moreno, Anita Bollmann-Giolai, Govind Chandra, Paul Brett, Jack Davies, Owen Thornton, Vinoy Ramachandran.

**Methodology:** Alba Pacheco-Moreno, Anita Bollmann-Giolai, Paul Brett, Philip Poole, Vinoy Ramachandran, James K. M. Brown, Paul Nicholson, Sarah DeVos, Jacob G. Malone.

**Project administration:** Jacob G. Malone.

**Resources:** Chris Ridout, Sarah DeVos.

**Supervision:** Philip Poole, James K. M. Brown, Paul Nicholson, Chris Ridout, Sarah DeVos, Jacob G. Malone.

**Writing – original draft:** Alba Pacheco-Moreno, Jacob G. Malone.

**Writing – review & editing:** Alba Pacheco-Moreno, Anita Bollmann-Giolai, James K. M. Brown, Paul Nicholson, Chris Ridout.

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
