## [Editor Report · Decision Letter 0]

21 Jun 2023

Dear Dr. Malone, 

Thank you for submitting your manuscript entitled "Barley cultivars shape the abundance, phenotype, genotype and gene expression of their associated microbiota by differential root exudate secretion" for consideration as a Research Article by PLOS Biology.

Your manuscript has now been evaluated by the PLOS Biology editorial staff, as well as by an academic editor with relevant expertise, and I am writing to let you know that we would like to send your submission out for external peer review.

Once your full submission is complete, your paper will undergo a series of checks in preparation for peer review. After your manuscript has passed the checks it will be sent out for review. To provide the metadata for your submission, please Login to Editorial Manager (https://www.editorialmanager.com/pbiology) within two working days, i.e. by Jun 23 2023 11:59PM.

Kind regards,

Paula

---

Senior Editor

PLOS Biology

---

## [Decision Letter · Decision Letter 1]

18 Sep 2023

Dear Dr. Malone,

Please allow me to first apologize for the delay in the processing of your manuscript. This delay is caused by my difficulty in recruiting reviewers for your manuscript during the summer holiday period. I am sorry for this, and I thank you for your patience while your manuscript "Barley cultivars shape the abundance, phenotype, genotype and gene expression of their associated microbiota by differential root exudate secretion" was peer-reviewed at PLOS Biology. It has now been evaluated by the PLOS Biology editors, an Academic Editor with relevant expertise, and by several independent reviewers. 

In light of the reviews, which you will find at the end of this email, we would like to invite you to revise the work to thoroughly address the reviewers' reports. Please thoroughly address all of the reviewer comments to strengthen the conclusiveness of the data.

As you will see below, the reviewers agree regarding the interest of the study and potential impact, but they all find some issues that would need to be solved. 

Given the extent of revision needed, we cannot make a decision about publication until we have seen the revised manuscript and your response to the reviewers' comments. Your revised manuscript is likely to be sent for further evaluation by all or a subset of the reviewers.

**IMPORTANT - SUBMITTING YOUR REVISION**

*Re-submission Checklist*

*Published Peer Review*

*PLOS Data Policy*

*Blot and Gel Data Policy*

Sincerely,

Paula

---

Senior Editor

PLOS Biology

REVIEWS:

Reviewer #1: Roots and microbes.

Reviewer #2: Genetic basis of plants' interactions with microbes.

Reviewer #1: In this elegant study, Pacheco-Moreno et al. study the microbiomes different barley cultivars assemble and aim to identify how Pseudomonas adapt to their host cultivar, the role of selected genes in rhizosphere competence and role of cultivar-specific microbiomes in plant performance. The study employs a range of well executed experiments and reaches some well-proven conclusions. Nevertheless, I have a number of comments that can be found below.

Major comments

1. In the figure preparation, I would prefer the bulk soil on the left side of the graph followed by the root or rhizosphere samples. Please take this into account in all figures about microbiome analysis.

2. Lines 208-217, what about fructose that displayed highest difference in the previous assay (luminescent biosensor)?

3. Lines 230-239, in previous experiment, axenic exudates were collected from the 2 cultivars and around 100 entities were found as differentially abundant. Did the authors consider observing growth of selected isolates on the "total" exudates, instead of testing specific carbon sources? I feel that it would be an extra confirmation about preferential growth of isolates on host-specific exudates.

4. Lines 309-323, considering that you generated a set of mutants adapted in each rhizosphere, it would be nice to also perform some growth assays with SBW25 and compare its effect to the mutants when they are applied in each cultivar. That will also show whether bacterial genes are not only linked with colonization but also with downstream effects such as growth.

Minor comments

1. Lines 115-123, please mention the figure that you are describing.

2. Lines 179-180, it would be nice to have info on isolates in a suppl. table.

3. Lines 194-198, it would benefit the reader to have this percentage in numbers in the text such as 13/45 isolates displayed trait x.

4. Line 226, 7/60 cannot be considered as several but as "only a few".

5. In Figure 4, good to add a dashed line at OD=0.3 since now the focus is on the dashed line that is lower than 0.3.

6. Figure 4e has low resolution. Replace with a higher quality one.

7. Figure 5a, it would be informative if the genes can be organized based on function and there is some info on the graph about general role (e.g. transporters, signaling, etc).

8. Lines 261-264, this is not what is shown in figure 5b.

9. Line 479, is it indeed 500 cm2 square plates?

10. In Materials and Methods, it should be Petri and not petri. Please also remove space between number and Celsius when mentioning temperatures (e.g. 4oC).

11. Line 701, it should be suppl. figure 5 and not 3. Please check all figure numbers in the text for correctness.

Reviewer #2: Author comments

In this manuscript Pacheco-Moreno et al. present evidence that two barley cultivars (one modern and one heirloom cultivar) exhibit differences in root exudate quality, which preferentially select different microbiomes & particularly different subsets of Pseudomonas fluorescens to their rhizospheres; and further, that the respective subsets of P. fluorescens show higher relative fitness on their preferred type of root exudates; that the differences in root exudate profiles are sufficient to alter gene expression of a reference strain of P. fluorescens; and finally that the subsets of P. fluorescens preferentially improve growth of the cultivar whose root exudates they prefer.

As a whole, if all of these conclusions are justified, then this would be perhaps the most complete demonstration (that I am aware of) of the causes and consequences of genetically-induced microbiome variation in a major crop plant. It would be remarkable to see a single piece of work that demonstrates an effect of host genotype on microbiome content, identifies the underlying trait, and confirms reciprocal fitness benefits for both host and microbe, as this manuscript proposes. So to my mind, the potential impact of the work is clearly high.

However, I have some questions and concerns about the strength of evidence for some of these claims.

All of the data came from experiments done in the lab or growth chamber. This approach has a lot of benefits, many of which are exemplified by this paper, but it comes with a fairly major drawback: a lack of generalizability to real-life field scenarios. This is not just an academic complaint- the difficulty of reproducing lab results in the field has been one of the primary obstacles preventing the deployment of very promising synthetic microbial communities for agricultural benefit. Now, I do appreciate the value of reductionist work like this, but I think it is important to address the downsides. The authors need to be up front about the setting of the work throughout the paper - i.e. the reader should not be left wondering until the methods section whether this was a field study. The authors also need to revise the Discussion to address the (extremely likely) possibility that host-microbiome dynamics are not so neat and tidy in the field. As part of this, there should be explicit discussion of how the presence of other, live microbes could modify the behavior and relative fitnesses of the cultivar-specific P. fluorescens strains. The Methods section indicate that even the most realistic part of the study — the amplicon sequencing data in the first section of the Results—came from plants grown in quite artificial conditions, i.e. a peat-based soilless potting media, which has been shown to harbor a microbiome that is largely unrepresentative of what is found in real soils. On that note I found it very odd that the authors chose to autoclave the field soil before mixing it with the potting media- without autoclaving, it could have been a rich inoculum of real soil microbiome. So in summary, while I enjoyed reading this paper and the story it told, I am skeptical that its main conclusions can be generalized to plant-microbe interactions in the real world.

The statistical support for some of the claims was questionable. In some cases, I simply did not understand the logic behind the choices of statistical tests. For example, in the first section of the Results, some microbiome comparisons used ANOSIM while others used ADONIS, with no reason given for the frequent switching between methods. Both are valid permutational MANOVA methods (though ADONIS is more robust), but why not just pick one and use it? Another unusual switching between methods is found in lines 158-164, where a Tukey post-hoc method was used for one alpha diversity metric and a very different approach - pairwise Wilcoxon test - was used for another metric. It must not be a matter of needing a non-parametric test due to problems meeting the assumptions for ANOVA, because in line 164 the ANOVA is used for both. Even if that were the justification, the Wilcoxon test is not an appropriate substitute for ANOVA followed by Tukey HSD: first, it can only be used to compare two groups whereas ANOVA can handle multiple groups; second, it should only be used on paired data, which (as far as I can tell) this experiment did not have. If there was some sort of pairing mechanism, it needs to be explained in the Methods.

In other cases, the statistical approach lacked sufficient detail. For example, simply reporting the p-value for an ANOVA tells me nothing about whether any particular explanatory variable was associated with the response variable (microbiome). To interpret ANOVA results, it is mandatory to specify exactly what explanatory variables, covariates, interactions, etc. were included in the model, as well as what method of calculating sums of squares was used (Type I, Type II, etc.). Different variables each have their own F statistic, df and p-value, which should be reported in the text as support instead of a p-value for the ANOVA in general. In simple cases (one-way ANOVA) they may be the same but as written the reader has no way to know that.

Finally, some claims had no statistical support at all, for example many of the comparisons of individual genera between cultivars or between compartments (lines 133-142, e.g.). Another example is the GC-MS data, and another is lines 215-220 where no evidence is given that the differences between Tipple and Chevallier exudates are strong enough to rule out whether it was just noise.

In summary, at least some of the data will need to be re-analyzed using appropriate statistical methods, and in general the Methods section should be expanded to provide adequate detail about what analyses were conducted. The authors should also be careful not to over-interpret their results when the statistical support is actually weak (for example, claims about the benefits of cultivar-selected microbiomes on plant growth).

Minor comments.

Line 67-68: I would not consider nodule formation to be a good representation of genotype effects on the rhizosphere more generally - nodulation is such a specific tightly defined interaction, whereas interactions in the rhizosphere are much more diffuse.

Lines 87-106: This is a long description of the results which I do not think belong in the introduction section.

Line 127: "Bacterial composition between cultivars" - is this referring to all 3 compartments together?

Figure 4: I find this figure kind of hard to interpret - is this really the clearest way to show these data? Why not just show the 48h timepoint or the maximum for each isolate?

Figure 2: Please write out the abbreviations in "Isolate phenotypes"

The caption for Figure 3b states that the heatmap shows metabolites with "a log2-fold change >= 2 in at least one cultivar". This is confusing to me, what is the log2 fold change in comparison to? Why not test for a log2 fold change between the cultivars?

Line 681: How much did the volumes need to be adjusted? Was the dilution the same for both rhizosphere suspensions? 

Line 646: Does this mean that 12 plants went in to each rhizosphere sample, or were 12 plants used total?

Lines 647-649: So each individual plant went into 20 mL liquid, or did multiple plants combine in one tube? Also, this is the first mention of "sample tubes containing vermiculite"

Methods section: Maybe I missed it but it seems that the "rooting solution" mentioned several times is never defined… what was in it?

Metabarcoding study: How did the authors contro

---

## [Decision Letter · Decision Letter 2]

17 Jan 2024

Dear Dr Malone,

Thank you for your patience while we considered your revised manuscript "Barley cultivars shape the abundance, phenotype, genotype and gene expression of their associated microbiota by differential root exudate secretion" for consideration as a Research Article at PLOS Biology. Your revised study has now been evaluated by the PLOS Biology editors, the Academic Editor and the original reviewers. 

You'll see that reviewer #1 says that “some issues still remain or not totally addressed” – most of these merely involve textual/presentational changes, but one request (their “major issue 2”) involves some additional experimental work, which the Academic Editor thinks should be straightforward. Reviewer #2 is mostly satisfied, but insists on you providing fuller information about their ANOVA model; their remaining points are requests for clarification.

In light of the reviews, which you will find at the end of this email, we are pleased to offer you the opportunity to address the remaining points from the reviewers in a revision that we anticipate should not take you very long. We will then assess your revised manuscript and your response to the reviewers' comments with our Academic Editor aiming to avoid further rounds of peer-review, although might need to consult with the reviewers, depending on the nature of the revisions.

**IMPORTANT - SUBMITTING YOUR REVISION**

*Resubmission Checklist*

*Published Peer Review*

*PLOS Data Policy*

*Blot and Gel Data Policy*

Sincerely,

Roli Roberts

Roland Roberts, PhD

Senior Editor

PLOS Biology

rroberts@plos.org

REVIEWERS' COMMENTS:

Reviewer #1:

The authors tried to address the concerns there were about the previous version and this is highly appreciated. Nevertheless, some issues still remain or not totally addressed. Please see below.

Major issues

1. It is probable that it would benefit the storyline if section "Chevallier and Tipple differ in root exudates composition" is presented first in the story. Then the authors can build their story on the fact that they have different exudation patterns, therefore they could differentially shape the microbiome. Then within the exudates, sugars have a pronounced effect on Pseudomonads, etc. 

2. Regarding section "Primary carbon metabolism is central to Pseudomonas spp. cultivar adaptation", I really think that testing the effect of exudate mix from each cultivar would be beneficial before testing individual metabolites. That will show indeed whether exudates of each cultivar can have an effect on Pseudomonas. If this is not possible for all isolates (which is understandable), a subset could be tested (with isolates deriving from the 2 cultivars). By subset, some of 42 isolates could be tested - line 233). I cannot see why this is that hard to do. I suspect that a big volume of exudate can be collected by barley and then test effects on Pseudomonas isolates in 96-well plates. That requires some ul per well.

3. Line 245, in their reply to comment on previous version, the authors mention that SBW25 is isolate from sugar beet, therefore its use here and any effects might be non specific for the host and not relevant for the cultivars used here. Then why to use SBW25 and its mutants?

4. Line 262, I still feel that info on function of the genes would benefit readability and function of figure. Now it's mostly showing how many isolates have a given gene, without knowing its function. Even a colored dot indicative of activity (e.g. transporter, secretion system, etc) on top of each bar could work.

5. Line 402, what about the effect on the plant (comment in previous version as well)? Does this difference or lack of in colonization have an influence on the effect of the bacteria on the plant? 

Minor issues

1. The title reads too general. Other than the first figure where the overall microbiome is assessed, in the rest of the manuscript the focus in on Pseudomonas sp. Therefore the authors should mention Pseudomonas and not microbiota in general.

2. Lines 84-85: The background research question is pretty general. How does this advance our knowledge from what Bulgarelli already showed also in Barley (lines 66-67)?

3. Lines 106-107: how is it possible that a microbiome is still present after sterilisation?

4. Line 131: Can 90% abundance of two genera be representative of field conditions, where more diversity is expected?

5. Line 161: observed richness? And df isn't it supposed to be number of replicates per group minus 1?

6. Line 168: in both cultivars Pseudomonas is among most abundant genera. Not sure about term "preferentially".

7. Line 175: please indicate how many are isolated from each cultivar.

8. Line 179: Pseudomonas is genus not species.

9. Line 197: please add statistics in Suppl. Figure S3.

10. Line 255, Illumina can be omitted.

11. Line 275, what the authors know is hexoses not specifically glucose, right? 

12. Lines 329-331: But why is it important to understand those ecological effects? What is the potential benefit? up to here it sounds like a mere curiosity.

13. Line 353, the authors should elaborate on why they focused on Pseudomonas before going into the details.

Reviewer #2:

I have now carefully reviewed the original submitted manuscript and this revised version. The authors have sufficiently addressed most of my concerns, with the following exception:

My original review stated, "To interpret ANOVA results, it is mandatory to specify exactly what explanatory variables, covariates, interactions, etc. were included in the model, as well as what method of calculating sums of squares was used (Type I, Type II, etc.). Different variables each have their own F statistic, df and p-value, which should be reported in the text as support instead of a p-value for the ANOVA in general. In simple cases (one-way ANOVA) they may be the same but as written the reader has no way to know that."

In lines 121-163 of the revised version, the authors provide some additional detail (e.g., the degrees of freedom) which only partially addresses my original comment. These paragraphs describe evidence for differences in microbiome composition and alpha diversity between various groups of samples. I still cannot fully interpret these results because the authors still do not provide the actual models being tested in the ANOVA(s). So let me clarify and repeat my request: the Methods section must include the statistical models in full, such as: 

"A two-way ANOVA with Type III sums of squares was used to model the alpha diversity metric as a function of Cultivar, Compartment, and the interaction between Cultivar and Compartment". Or, "A one-way ANOVA with Type III sums of squares was used to model alpha diversity, with plant Compartment as the explanatory variable; data from both cultivars were pooled for this analysis". Any given F-test for the Compartment effect (for example) would be interpreted differently depending on which of those two models it came from, because the first approach simultaneously controls for the Cultivar effect and allows the Compartment effect to vary between cultivars; in contrast, the second approach simply reports the Compartment effect averaged across all levels of all other variables involved in the study. So, this information is mandatory and must be provided in the Methods for each ANOVA, Kruskal-Wallis, ANOSIM, and ADONIS test that was performed.

A few other minor comments:

Line 92: Is it appropriate to say that genotype-driven selection in the rhizosphere is "primarily" driven by root exudates when alternative mechanisms were not really explored?

Lines 109-111: These add up to 100%, which indicates that there were not any microbiome members from phyla other than these four. This is unusual for a plant microbiome study. What could explain the total absence of organisms from other phyla commonly seen in the rhizosphere, such as Firmicutes/Bacillota and Verrucomicrobiota? 

Lines 151-153: What is the Krustal test? Perhaps meant to be Kruskal-Wallis?

Lines 218-219: The use of "log2 fold change" and "log2 abundance" here is quite confusing. When describing the difference between cultivars, the authors report a "log2 fold change" for each of the two cultivars, which implies that the exudate concentration in each cultivar was compared to some shared control or standard, which is not mentioned. In my original review I had asked a similar question about Fig. 3b, to which you confirmed that the log2 fold change *between cultivars* was calculated directly, rather than using a comparison to a shared control or standard. So I am struggling to understand what these two reported log2 fold changes for the cultivars (16.474 for Chevallier, and -12.613 for Tipple) actually are. Is it possible that these numbers are actually the log2-transformed abundances (not fold changes), like what you reported for D-glucose in line 220? I am further confused because the "derivative of phosphoric acid" was described as absent from Tipple, so how was this 0 value transformed into a log2 fold change? Please clarify what these numbers actually are.

---

## [Editor Report · Decision Letter 3]

29 Feb 2024

Dear Dr Malone,

Thank you for your patience while we considered your revised manuscript "Barley cultivars shape the abundance, phenotype, genotype and gene expression of their associated microbiota by differential root exudate secretion" for publication as a Research Article at PLOS Biology. This revised version of your manuscript has been evaluated by the PLOS Biology editors and the Academic Editor.

Based on our Academic Editor's assessment of your revision, we are likely to accept this manuscript for publication, provided you satisfactorily address the following data and other policy-related requests.

IMPORTANT - Please attend to the following:

a) Please change the Title to a more concise version; we suggest "The genotype of barley cultivars influences multiple aspects of their associated microbiota via differential root exudate secretion"

b) Please mention the species name of the host plant (Hordeum vulgare, I assume?) in the Abstract.

c) Please address my Data Policy requests below; specifically, we need you to supply the numerical values underlying Figs 1ABCD, 2 (treefile), 3AB, 4ABCDE, 5AB, 6ABCDEFGH, 7, S1ABCD, S2ABCD, S3ABCDEFGHIJKL, S6ABCDEFG, either as a supplementary data file or as a permanent DOI’d deposition.

d) Please cite the location of the data clearly in all relevant main and supplementary Figure legends, e.g. “The data underlying this Figure can be found in S1 Data” or “The data underlying this Figure can be found in https://doi.org/10.5281/zenodo.XXXXX”

e) Please make any custom code available, either as a supplementary file or as part of your data deposition.

We expect to receive your revised manuscript within two weeks. 

*Published Peer Review History*

*Press*

Sincerely,

Roli Roberts

Roland Roberts, PhD

Senior Editor

rroberts@plos.org

PLOS Biology

DATA POLICY:

Regardless of the method selected, please ensure that you provide the individual numerical values that underlie the summary data displayed in the following figure panels as they are essential for readers to assess your analysis and to reproduce it: Figs 1ABCD, 2 (treefile), 3AB, 4ABCDE, 5AB, 6ABCDEFGH, 7, S1ABCD, S2ABCD, S3ABCDEFGHIJKL, S6ABCDEFG. NOTE: the numerical data provided should include all replicates AND the way in which the plotted mean and errors were derived (it should not present only the mean/average values).

CODE POLICY

Per journal policy, as the code that you have generated is important to support the conclusions of your manuscript, we require that you make it available without restrictions upon publication. Please ensure that the code is sufficiently well documented and reusable, and that your Data Statement in the Editorial Manager submission system accurately describes where your code can be found.

SPECIES INDICATED IN THE ABSTRACT? 

- Please note that per journal policy, the model system/species studied should be clearly stated in the abstract of your manuscript. 

We require the original, uncropped and minimally adjusted images supporting all blot and gel results reported in an article's figures or Supporting Information files. We will require these files before a manuscript can be accepted so please prepare and upload them now. Please carefully read our guidelines for how to prepare and upload this data: https://journals.plos.org/plosbiology/s/figures#loc-blot-and-gel-reporting-requirements

DATA NOT SHOWN?

---

## [Editor Report · Decision Letter 4]

19 Mar 2024

Dear Dr Malone,

Thank you for the submission of your revised Research Article "The genotype of barley cultivars influences multiple aspects of their associated microbiota via differential root exudate secretion" for publication in PLOS Biology. On behalf of my colleagues and the Academic Editor, Cara Haney, I'm pleased to say that we can in principle accept your manuscript for publication, provided you address any remaining formatting and reporting issues. These will be detailed in an email you should receive within 2-3 business days from our colleagues in the journal operations team; no action is required from you until then. Please note that we will not be able to formally accept your manuscript and schedule it for publication until you have completed any requested changes.

Sincerely, 

Roli Roberts

Senior Editor

PLOS Biology

rroberts@plos.org